# Simulating the effects of sea level rise and soil salinization on adaptation and migration decisions in Mozambique

Kushagra Pandey[1*], Jens A. de Bruijn[1,2], Hans de Moel[1], W.J. Wouter Botzen[1], Jeroen C.J.H. Aerts[1,3]

[1]Institute for Environmental Studies, VU Amsterdam, The Netherlands
[2]International Institute for Applied Systems Analysis (IIASA), Laxenburg, Austria
[3]Deltares, Delft, The Netherlands

*Correspondence to*: Kushagra Pandey (k.pandey@vu.nl)

**Abstract.** Coastal flooding and sea level rise (SLR) will affect farmers in coastal areas, as increasing salinity levels will reduce crop yields, leading to a loss of net annual income for farming communities. In response, farmers can take various actions. To assess such responses under SLR, we applied an agent-based model (ABM) to simulate the adaptation and migration decisions of farmers in coastal Mozambique. The ABM is coupled with a salinization module to simulate the relationship between soil salinity and SLR. The decision rules in the model (DYNAMO-M) are based on the economic theory of subjective expected utility. This theory posits that households can maximize their welfare by deciding whether to (a) stay and face losses from salinization and flooding, (b) stay and adapt (e.g., switching to salt-tolerant crops and enhancing physical resilience such as elevating houses), or (c) migrate to safer inland areas. The results show that coastal farmers in Mozambique face total losses of up to US$12.5 million per year from salt intrusion and up to US$1,200 million per year from flooding of buildings (RCP8.5 in the year 2080). Sorghum farmers may experience little damage from salt intrusion, while rice farmers may experience losses of up to US$4,000 per year. We show that medium-sized farmers (1–5 ha) are most at risk. This is because their farm size means that adaptation costs are substantial, while their incomes are too low to cover these costs. The number of households adapting varies between different districts (15%–21%), with salt adaptation being the most common, as costs are lowest. Despite adaptation measures, about 13%–20% of the total 350,000 farmers in coastal flood zones will migrate to safer areas under different settings of adaptive behaviour and different climatic and socioeconomic scenarios.

## 1 Introduction

With climate change and rising sea levels, coastal communities will increasingly face the risk of flooding, affecting their livelihoods. In addition, sea level rise (SLR) will further increase the salinization of coastal agricultural lands, impacting the fertility of coastal soils and crop yields (Materechera, 2011; Montcho et al., 2021). With an economy that is 70% dependent on agriculture (World Bank, 2017) and two-thirds of the population living in coastal areas, Mozambique already suffers from flooding and salinization in coastal zones. Given the projected trends, Mozambique is investigating adaptation options for coastal farmers to reduce the risk associated with SLR. Measures such as switching to salt-tolerant crop varieties may help

farmers who have the resources and capacity to implement adaptation measures. For others, however, migration to safer locations may become inevitable (Nicholls and Cazenave, 2010). Several studies have been conducted on the impact of salinity on crop production in Mozambique and other regions. For example, the Instituto de Investigação Agrária de Moçambique (IIAM) conducted an initial study that identified salinity as a problem, with Electrical conductivity (ECe) values exceeding 16 dS m-1 in coastal areas. Additional data were published in subsequent studies at the national level (e.g., FAO & ISRIC, 2012) and at the global level (e.g., Ivushkin et al., 2019; Hassani et al., 2020; Hassani et al., 2021). Hassani et al. (2020) simulated salinity maps in agricultural areas and estimated 85,350 ha of salt-affected area in Mozambique. In addition, several studies have assessed how increased salinity levels may affect crop yields at different scales. For example, estimates based on remote sensing show that salt stress in plants limits their ability to take up water (Ivushkin et al., 2019; Madrigal et al., 2003). As a result, saline soils can reduce the fertility of arable land and decrease yields by more than 50% (Anami et al., 2020; Ivushkin et al., 2019). FAO (2021) published a map showing the spatial distribution of salt-affected areas in Mozambique as highly saline. Furthermore, Hasegawa et al. (2022) project the impact of climate change on crop yields in a global dataset reviewing 202 studies from 1984 to 2020 in 91 countries. They also consider the adaptation options of fertilizers, irrigation, cultivars, soil, organic matter management, planting time, and tillage with irrigation and fertilizers as the most important adaptation options. Adaptation by changing crop type can increase crop yield by 7%–15% (Challinor et al., 2014). While SLR-induced migration in coastal areas has received attention in recent years (Reimann et al., 2023; Hauer et al., 2020), these studies mostly focus on migration related to flood risk. There are currently only a few studies on the effects of salinization and SLR on migration. For example, Chen and Mueller (2018) studied coastal Bangladesh and used a regression approach to observe migration to inland areas. Duc Tran et al. (2023) interviewed farmers in coastal provinces of Vietnam's Mekong Delta and assessed the perspectives of 120 farmers on rural out-migration. They found that rural out-migration is closely related to household vulnerability to natural disasters such as drought and salt intrusion in the Mekong region. In addition, the dynamic interactive vulnerability assessment (DIVA) model (Vafeidis et al., 2008; Hinkel and Klein, 2009) is a widely used modelling framework for studying coastal systems, including coastal erosion, coastal flooding, and salt intrusion in deltas and estuaries (Wolff et al., 2016; Fang et al., 2020). However, DIVA does not account for salinity intrusion into coastal aquifers. No studies have assessed the combined effects of flooding and salinization on both adaptation and migration responses in coastal areas. In order to simulate the effects of SLR and salinization on the migration of coastal farmers, a model is needed that can simulate adaptation and migration decisions (and the trade-offs between them) under different scenarios of future salinization. Several methods could be used to address this challenge. For example, statistical models (e.g., Chen and Mueller, 2018) can be useful but often require large amounts of data to produce significant results and may therefore be less suitable for a data-poor region such as Mozambique. Furthermore, a popular simulation model for migration and climate change is the gravity model (Cameron, 2018; Mallick & Siddiqui, 2015; Robinson et al., 2020; Simini et al., 2012). This model uses distance and population size (in the origin and destination of migrants) as the main drivers of migration (Lee, 1966). Gravity models are useful tools for exploring aggregated migration flows but cannot be used for individual adaptation and migration decision making, where individual decisions are highly dependent on household characteristics, assets, and the environment. Agent-based models

(ABMs) help to combine insights from different disciplines by addressing the role of bounded rationality, social interaction, and agent heterogeneity (Savin et al., 2023). They allow studying complex systems involving behaviour, economics, psychology, agriculture, and climate (Castro et al., 2020). ABMs are increasingly used for adaptation modelling (Haer et al., 2020; de Ruig et al., 2022; Streefkerk et al., 2023; De Bruijn et al., 2023). Klabunde et al. (2016) have conducted a review study that suggests using different behaviour theories within ABMs for migration. As we focus here on individual farmers' decisions on adaptation and migration, ABMs have emerged as promising tools (Thober et al., 2018). ABMs allow us to assess how individual farmers' decisions are influenced not only by their environmental context (flooding, salinization, etc.) but also by other agents, such as the government. For example, Cai and Oppenheimer (2013) used an ABM to simulate climate-induced agricultural labor migration in the United States. Another recent example is the DYNAMO-M model built for France. This model simulates household decisions in response to coastal flooding by evaluating trade-offs between adaptation and migration. Such decisions are made under different scenarios of SLR, flooding, and government intervention for flood protection (Tierolf et al., 2023). However, this model focuses only on the flood adaptation of households in urban areas and does not include salinization processes or the impact on crop yields for rural farmers. The main goal of this paper is to investigate the interlinked migration and adaptation responses to both flood risk and salinization. This study is the first to develop a model to simulate both the adaptation and migration decisions of farmers in Mozambique under different SLR and salinization scenarios. We further improve the model by adding a novel database of household characteristics. We run the model with an annual time step from the current year to 2080 and also include flood risk as a second environmental driver alongside salinization. The remainder of this paper is organized as follows: Section 2 discusses the case study; Section 3 describes the methods, including the ABM and data; and Section 4 presents the modelling results. Section 5 and 6 discuss the results, limitations, and conclusions, respectively.

## 2 Case Study: Mozambique

Mozambique is a coastal country in southeastern Africa with a population of 33 million, nearly 70% of whom work in agriculture. With a 2,470 km coastline along the Indian Ocean and tens of thousands of people living in coastal floodplains, the country faces a high risk of coastal flooding from tropical cyclones (Neumann et al., 2015).

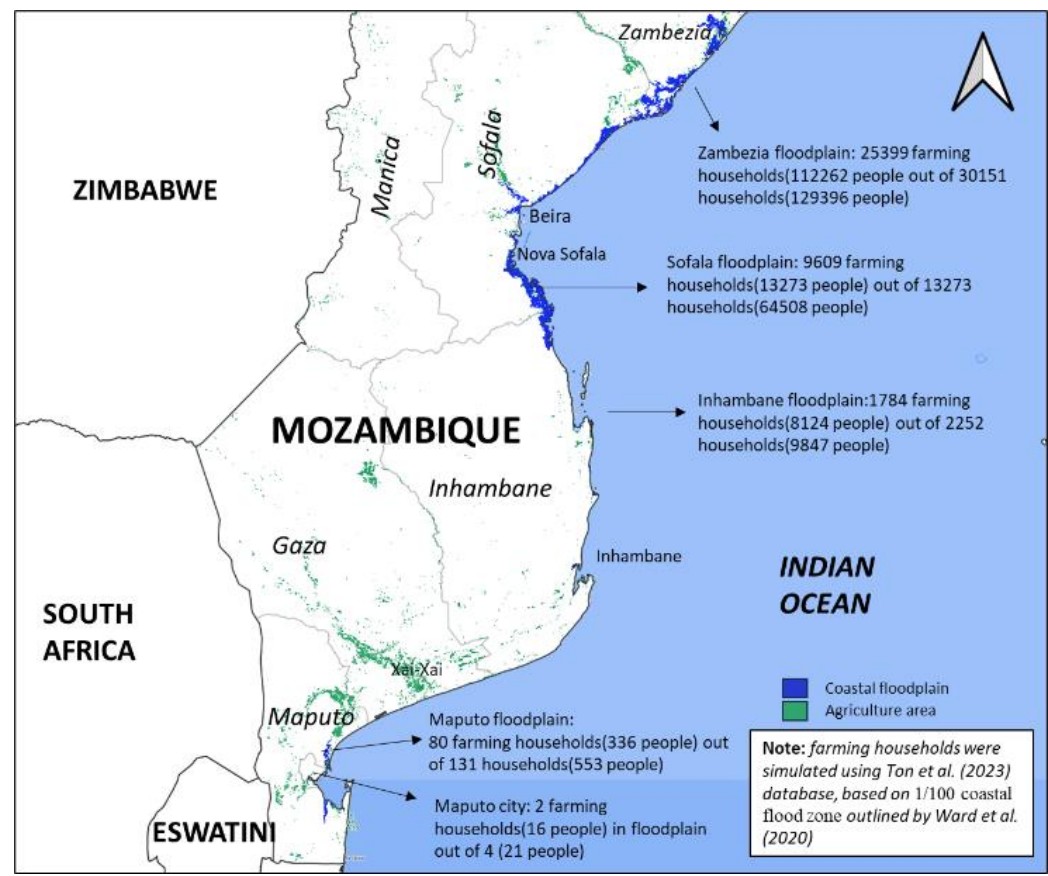

**Figure 1** Map of the coastal flood zones in Mozambique and number of households in the year 2015 (source: Ton et al., 2023).

Mozambique has experienced several floods in the last decade. For example, the recent cyclone Idai flood in 2019 affected 3 million people, with 1.85 million in Mozambique (Relief web 2019). The impact was huge, with 905 fatalities and an estimated economic loss of US$3 billion (Nhundu, 2021). Figure 1 shows the flood zones affected by coastal flooding only, based on Ward et al. (2020). Figure 1 also shows the number of households involved in farming in the flood zone using a database from Ton et al. (2023). In total, 48,651 farming households (219,194 farmers) live in coastal floodplains. In addition to the direct effects of flooding on buildings and infrastructure, agriculture will increasingly be affected by salt intrusion and lower yields. These impacts may affect the entire economy since 64% of Mozambique's total land area is agricultural, and 27% of the GDP comes from agricultural exports (The World Bank, 2017). The harvested area includes 47% rice, 26% maize, 16% cassava, and 11% legumes (see Supplementary annex S1.1).

Mozambique's dependence on agricultural exports also makes it one of the most vulnerable and least prepared countries for climate change-related risks (UND, 2015). For example, salt intrusion reduces yields, as most farmers in Mozambique grow rice, which is not a salt-tolerant crop. The relatively low GDP per capita of $514.5 in the year 2022 (World Bank 2022) makes

105    it difficult for households to adapt. As a result, Mozambique is also the third-largest recipient of climate finance, receiving around $147.3 million in 2016 (HBS, 2016).

There are several adaptive responses to reduce climate risk: (1) At the farm level, farmers can reduce excess salt levels in the soil by applying irrigation, using manure or compost (Islam et al., 2017), adding gypsum, or applying topsoil replacement (Ibrahim et al., 2012; Sarwar et al., 2011; Tahir & Sarwar, 2013). In addition, a generally accepted and sustainable adaptation

measure is the use of a salt-tolerant crop variety (Atzori, 2022; Bourhim et al., 2022; Negacz et al., 2022). (2) In order to reduce the direct impact of flooding on assets and people, the government is currently assessing flood risk and investing in flood risk management, such as levees. Although Mozambique does not have national flood protection standards, new flood adaptation plans are being implemented to protect people and assets, for example, around the city of Beira. However, most government projects focus on the population in urban centres and often exclude the rural population. Rural households are

mostly dependent on individual flood adaptation measures, such as raising houses. (3) If climate adaptation measures, either by the government or by individual households and farmers, fail, people may have no choice but to leave the affected low-lying areas (Fion De Vletter, 2007). Internal socioeconomic-driven migration has already been an issue in Mozambique since the 1980s (First, 1983) apart from out-migration to other south African countries (Facchini et al., 2013), and has led to internal migration from the poor rural south to the northern cities in search of better employment opportunities. SLR, increased flooding

and land degradation due to salinization may further trigger migration from the coast to safer areas.

### 3 Methods

Figure 2 shows how we extend the DYNAMO-M ABM of Tierolf et al. (2023) with a salt intrusion module. The ABM simulates household migration and adaptation decisions based on the discounted expected utility (DEU) theory. These

decisions are tested under SLR (RCPs 4.5 and 8.5) and socioeconomic development (SSP2) over the period 2020–2080 with annual time steps. While the ABM simulates the adaptation and migration behaviour of households living in the 1/100 coastal flood zone, a coupled gravity-based migration model simulates internal migration flows towards the coastal flood zone and between inland areas (departments). At each annual timestep, farmers can (a) reduce flood risk by implementing floodproofing measures to protect their homes and (b) reduce soil salinity risk on their farmland by switching to a more salt-tolerant variety.

These decisions are influenced not only by the level of salinization and flood risk but also by the socioeconomic characteristics of farming households and their farm size based on available statistics. Every year, soil salinity increases due to a steady SLR (Figure 3). In addition, a flood event may occur each year within the flood zones with a probability associated with return periods of up to 1,000 years (i.e., the flood associated with a 5-year return period has a 20% probability of occurring each year). If such a flood occurs, the salinity of the soil will also increase due to the salt deposited by the flood water.

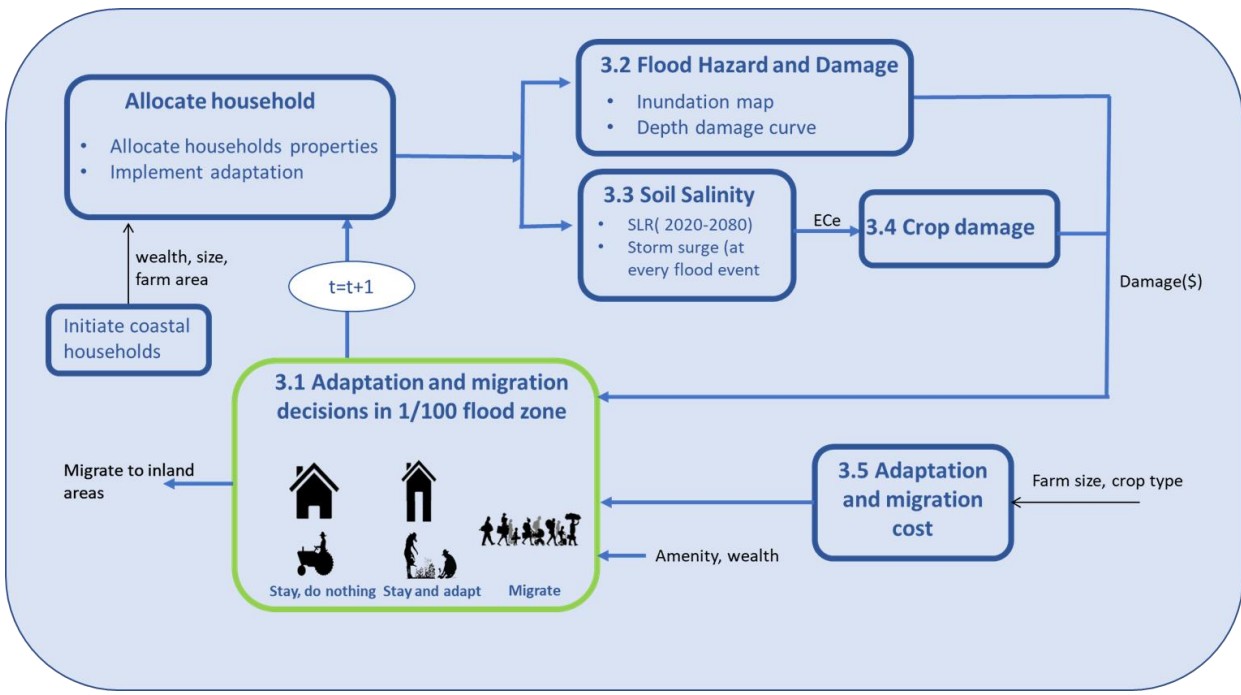

**Figure 2.** Modelling framework. We expand the DYNAMO-M model (Tierolf et al., 2023) by addressing the relationship between salinity intrusion and farmers' adaptation and migration decisions. We focus on households in the 1/100 flood zone that make boundedly rational adaptation and migration decisions based on the discounted expected utility (DEU) theory.

3.1 Adaptation and migration decisions in the 1/100 flood zone

Before running the model, we first generate household agents and their key socio-economic characteristics (income, education, age) that are statistically similar to the actual population in the 1/100 flood zone using GLOPOP-S database (Ton et al., 2023). Each farmer is assigned a farm of a certain size. The farm size is important because it determines the potential yield, damage, and income of a farmer. Therefore, each farmer is initially assigned a farm size based on probability distributions of statistical

information on farm sizes per district based on Lowder et al. (2016; see Supplementary 1.3 for details). Natural population change and GDP growth are based on population change rates available for all departments in 2016 and a medium population growth scenario based on SPP2 (see Supplementary information S1.2).

Migration and adaptation decisions of households in the 1/100 flood zone follow the DEU theory (Fishburn et al., 1981). This method allows households to weigh adaptation options against migration, taking into account the costs and benefits of

adaptation and migration, as well as risk perceptions and preferences related to their experience of flood risk. The model runs from 2020 to 2080, with annual time steps. In each time step, households maximize their DEU according to the following decisions:

- Do nothing (Eq. 1).
- Implement elevating measures and adapt to salt intrusion (Eq. 2).

- Migrate to another region $y$ (Eq. 3).

$$DEU_1 = \int_{p_i}^{p_I} \beta_t * pi * U\left(\sum_{t=0}^{T} \frac{W_t + A_{x,t} + Inc_{x,t} - D_{x,t,i}}{(1+r)^t}\right) dp \qquad \text{(Eq. 1)}$$

$$DEU_2 = \int_{p_i}^{p_I} \beta_t * p_i * U\left(\sum_{t=0}^{T} \frac{W_t + A_{x,t} + Inc_{x,t} - D_{x,t,i}^{adapt} - C_t^{adapt}}{(1+r)^t}\right) dp \qquad \text{(Eq. 2)}$$

$$DEU_3 = U\left(\sum_{t=0}^{T} \frac{W_t + A_{y,t} + Inc_{y,t} - C_{y,t}^{migration}}{(1+r)^t}\right) \qquad \text{(Eq. 3)}$$

In these equations, DEU is a function of $W_t$ (wealth), $A_{x,t}$ (amenities: e.g. the value of living near water) in region $x$ and $A_{y,t}$ in region $y$, $Inc_{x,t}$ (income) in region $x$ and $Inc_{y,t}$ in region $y$, $D_{x,t,i}$ (flood damage to buildings + salt damage to crops), $C_t^{adapt}$ (costs of adaptation to both flood and salt intrusion), and $C_{y,t}^{migration}$ (costs of migration).[1] We apply a time discounting factor $r$ of 3.2% (Evans & Sezer, 2005) over a time horizon (T) of 15 years, which is the number of years a homeowner stays in their home on average.[2] These Utilities are added over the flood events (differentiated by return-period) where $p_i$ is the exceedance probability of these flooding events and $\beta_t$ is risk perception at a time t. We refer to De Ruig et al. (2022) and Tierolf et al. (2023) for the values of the risk perception parameter $\beta_t$ and the risk aversion parameter of the utility function $U$. The DEU

---

[1] Our formulation of the discounted expected utility functions includes a summation of monetary terms that occur over time as is line with related ABM applications (e.g. Haer et al., 2019; de Ruig et al., 2022; Tierolf et al., 2023), instead of a summation of discounted utility values themselves over time (Coble and Lusk, 2010). Our approach is consistent with the use of a time discount rate estimated for monetary values instead of a utility discount rate, but may be a simplification for capturing agents' preferences related to the temporal distribution of the included monetary amounts over time. Although we do not have data on such preferences for Mozambique to directly tests for this, the model calibration and validation exercises show that our behavioural rules adequately predict observed adaptation decisions in Mozambique (see sections 3.7 and 4.2). This gives confidence in our approach.

[2] This value of the time discount rate is based on estimates derived from the European context, since a Mozambique estimate is lacking. One could expect that the actual discount rate in Mozambique is higher than this value, resulting in a too high weight given to monetary values in the far future. However, such an effect is counteracted by our choice for a relatively short time horizon of 15 years over which future values are included in the utility calculation. Our model calibration and validated analyses demonstrate that our combined choice of behavioural parameters performs well, in a sense that modelled adaptation outcomes match those observed in Mozambique with survey data (see sections 3.7 and 4.2).

functions in Equations 1, 2, and 3 are a function of risk aversion ($\sigma$) (Equation 4), which is assumed to be constant based on Gandelman, Nestor (2015).

$$U(x) = \frac{x^{1-\sigma}}{1-\sigma} \qquad \text{(Eq. 4)}$$

The risk perception parameter $\beta$ is used to capture bounded rationality. Following Ruig et al. (2022) and Haer et al. (2020), we define $\beta$ over t years, as in Equation 5, where $c$ and $d$ are constants:

$$\beta_t = c * 1.6^{-d*t} + 0.01 \qquad \text{(Eq. 5)}$$

If households decide to migrate away from the flood zone, they can move to other inland departments or to another coastal department with a flood zone.

## 3.2 Flood hazard and damage

We assume a general coastal protection standard in Mozambique of 1/10 years for all coastal areas and exclude higher return periods (1/2 years and 1/5 years) from our analysis (Scussolini et al., 2016). The simulated flood level at each household's geographical location is based on a range of return periods, including once every 20, 50, 100, 200, 500 and 1,000 years, as shown on coastal flood maps produced by the AQUEDUCT flood analyzer framework (Ward et al., 2020). We interpolate between historical and projected flood levels in 2030, 2050, and 2080 to derive annual inundation levels under RCPs 4.5 and 8.5 (van Vuuren et al., 2011). Following Tierolf et al.'s (2023) method, each household samples inundation levels for all return periods for their current location in the floodplain based on the selected climate change scenario. Synthetic future flood events are simulated by randomly selecting an event type (by return-period) for each administrative unit and the exceedance probability of each flood event (e.g. a 1/ 10-year flood has a 10% chance of occurring at each time step; a 1/ 200-year flood has a 0.5% chance, etc.). A maximum damage value specific to Mozambique is used together with depth damage curves for residential structures to determine flood damage as a function of inundation level, which were both obtained from Huizinga et al. (2017). Dry flood proofing measures prevent water from entering the structure. This is captured by modifying the depth damage curves such that damage is reduced by 85% for water levels below 1 m (De Ruig et al., 2022). Inundation above 1 m overcomes the dry flood proofing, resulting in complete damage. A study by the World Bank in 2000 on flooding in Mozambique estimated that construction costs in rural Mozambique are six times lower than those of urban houses, while rural houses are half the size of urban houses. Therefore, we reduce the property value in rural areas by a factor of 3, so that the difference between urban and rural areas is better captured.

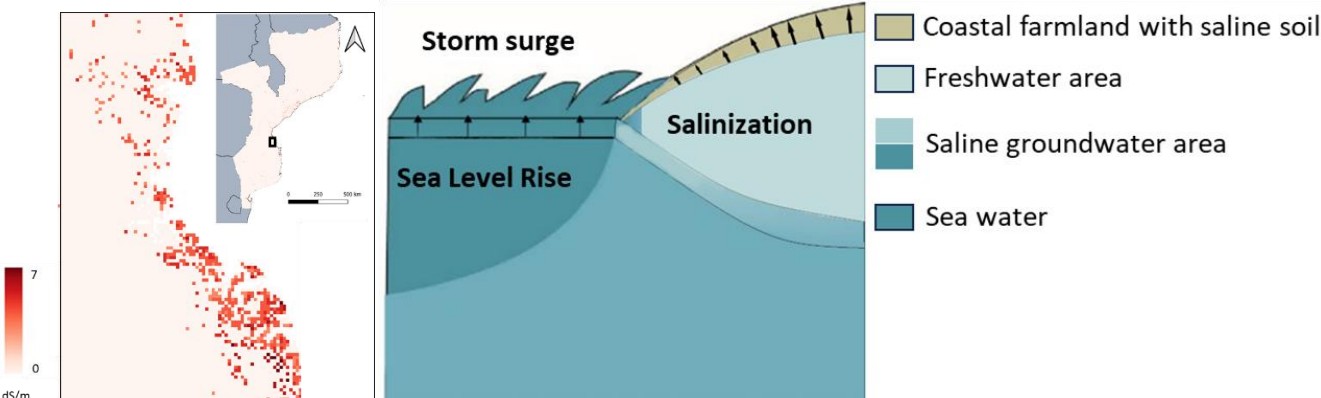

**Figure 3.** *Left:* soil salinity in dS/m for the year 2015 in coastal Mozambique based on Hassani et al. (2020). *Right*: conceptual diagram of salt intrusion processes adapted from Klassen and Allen (2017). Here, soil salinity is influenced by SLR and the synthetic storm surges.

### 3.3 Soil salinity

*Initial soil salinity:* Figure 3 shows the initial soil salinity values (dS/m) at the beginning of the simulation, which is simulated using data from Hassani et al. (2020; see Supplementary 1.3 for details).

*Future soil salinity:* We assume that topsoil salinity is only affected by two processes: (a) a gradual increase in salinity due to SLR (e.g. increased saltwater intrusion) and (b) flooding events (see Section 3.2), which increases the salinity levels after a flooding event. The two values are then added together to obtain the total salt deposition in the topsoil at the end of the year.

The geographical location of the farm determines the relative influence of SLR and flood events on the salinity levels of the farm. In the simulations, the initial salinity map is updated at each annual time step, following Klassen and Allen's (2017) concept of salt intrusion processes (Figure 3, right). We conceptualized the two salinization processes as follows:

(a) *Soil salinity due to SLR*: We use the latest available global soil salinity map from Hassani et al. (2020) as our baseline map in 2015. We extrapolated these values to 2080 under SLR scenarios, assuming increases of 50% and 100% for

RCP4.5 and RCP8.5, respectively, all based on Hassani et al. (2021). These assumed increases in salt intrusion are interpolated for each time step of the model. Figure 4 shows the soil salinity under RCP4.5.

(b) *Soil salinity due to flood events*: Increases in salinity levels can also be caused by saltwater flooding of land (Taylor & Krüger, 2019). To simulate this effect during a flood event, we assume that salt accumulates in the top layer of the soil ($EC_{e,soil}$) according to Equation 6. The total salt level (measured in dS) in the topsoil layer depends on the farm

size $A_i$. In order to calculate the increased salinity levels, it is assumed that all the salt from a flood event is absorbed by the top 1 m soil layer, using flood depths to calculate volume from Ward et al. (2020).

$$EC_{e,soil}^{flood} = \sum_{i=1}^{n} \frac{WL_i * A_i * EC_{e,sea}}{V_{farm}} \qquad \text{Eq. (6)}$$

Where *n* represents the total number of farming households in the coastal region, $WL_i$ is the flood water depth at the farm location, $A_i$ is the farm area, $EC_{e,sea}$ is the sea surface salinity (Boutin et al., 2021), and $V_{farm}$ is the volume of

215 affected soil, considering the root depth as 1 m.

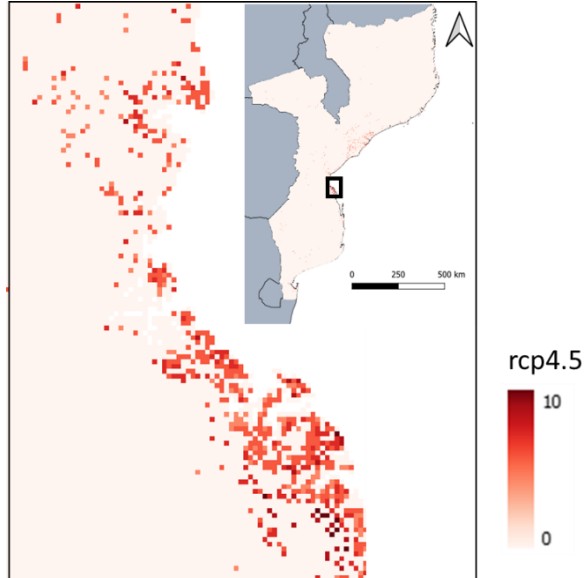

**Figure 4** Soil salinity for an area in central Mozambique (small black square) in the year 2080 under RCP4.5 SLR scenarios

### 3.4 Crop damage

Using the updated salinity levels from Section 3.3, we can calculate the annual salinity damage to crop yields. To do this, we first initialize the model by assigning one of the four dominant crops to a farm: rice, maize, sorghum or cassava (see Supplementary material S1.3 for details). These four crops account for 98% of the cultivated land in Mozambique (World Bank, 2017). The agricultural map for some crops is shown in Figure S1.

Next, we will convert the annual salinity levels into losses per crop yield type $Y_r$, following Maas and Hoffman (1977; Equation 7):

$$Y_r = 100 - b(EC_e - a) \qquad \text{(Eq. 7)}$$

Where $Y_r$ is the percentage crop yield loss relative to an optimal yield $Y$, $EC_e$ is the predicted electrical conductivity expressing soil salinity, and the constants $a$ and $b$ are crop-dependent parameters. $a$ is the threshold at which crop yield begins to deteriorate, and $b$ is the rate of deterioration type and is regularly updated by FAO (FAO 2002, Table 23).

Next, we convert the percentage yield losses $Y_r$ into monetary damages per farm using a damage function (Equation 8):

$$D = \sum_{j,k=0}^{j=n,k=3} Y_r(k) * Y(j,k) * A(j,k) * P(k) \qquad \text{(Eq. 8)}$$

Where $k$ is the crop index for the four crops, $j$ is the farm index. For each value of $k$, $Y_r$ is the relative yield as compared to $Y$, the current yield. $A$ is the individual farm size, and $P$ is the selling price. We use the spatial distribution of yields across existing farms in Mozambique from the GAEZ v4 portal (https://gaez.fao.org/). Farm sizes $A$ for different farmers are simulated based on a farm size distribution following Lowder et al. (2016). Using the producer prices for different crops from FAO (2015) and farm size, we can calculate damages to farming households.

Due to salt intrusion, farmers can adapt with one measure: switch to a salt-tolerant variety of their crop if the projected damage *after* the adaptation ($D_{x,t,i}^{adapt}$) is lower than without adaptation, all subject to the salt tolerance parameters in Equation 7. Van Straten et al. (2021) conducted field trials on salt-tolerant varieties of potatoes and observed that salt tolerance can increase up to twofold, and the rate of deterioration is reduced by half. Salt farm foundation (2016) observed similar factors in their field trials with six other crops. Therefore, we considered the same factors for four crops commonly used in Mozambique (see Table 1). These coefficients are then used in Equation 7 to calculate $D_{x,t,i}^{adapt}$.

## 3.5 Adaptation and migration costs

### 3.5.1 Adaptation cost

The variable $C_t^{adapt}$ in Equation 2 is the total adaptation cost for a household in a given year and is the sum of the cost of elevating the house $C_{annual}^{building}$ and the cost of crop adaptation $C_{annual}^{crop}$.

*Cost of flood adaptation:* $C_{annual}^{building}$: Households can floodproof their homes by elevating them. In determining the cost of adaptation $C_{annual}^{building}$, we used a fixed cost $C_0^{building}$ of \$1,861 per building at a fixed interest rate $r$ (World Bank, 2022) and loan duration $n$ as in Equation (9). These fixed costs are considered a fixed proportion of the property value, as calculated by Hudson (2020) and Huizinga et al. (2017). Aerts (2018) estimates similar values for other developing countries, such as Bangladesh and Vietnam. Based on World Bank (2000), similar to property values, we assumed that adaptation cost for rural households is three times lower than for urban houses.

$$C_{annual}^{building} = C_0^{building} * \frac{r*(1+r)^n}{(1+r)^n - 1} \qquad \text{(Eq. 9)}$$

*Cost of crop adaptation:* In order to reduce the impact of salinization, farmers can switch to a salt-tolerant crop variety (for four crop types, see Table 1). The cost associated with switching a crop is represented by $C_{annual}^{crop}$. The decision to switch to a salt-tolerant variety depends on the crop itself and other parameters, such as exposure to previous risks. However, for simplicity, we assume that this is the cost quoted by seed companies. Seed Co., founded in Zimbabwe, has testing and production sites in Mozambique, so we use their prices as a proxy for crop switching (*Mozambique - Access to Seeds*, 2019). The seed cost per hectare is calculated using the seed requirement (kg) per hectare of 25 kg/ha (*Crop Production Guidelines*, 2017) and the seed cost (\$/kg) from a local seed company (SC 419 - Seed Co. Zimbabwe Online Shop, 2023). By multiplying the seed cost (\$/kg) by the seed requirement (kg/ha) and farm size (ha), the total crop adaptation cost per farm can be calculated in US dollars.

**Table 1 Regular and salt-tolerant crops in Mozambique and their salt-tolerant varieties**

| Crop type | Regular variety | Salt-tolerant variety | Costs |
| --- | --- | --- | --- |

|  | Crop yield (t/ha) | Tolerance threshold (a) | Rate of deterioration (b) | Tolerance threshold (a: Eq 7) | Rate of deterioration (b; Eq 7) | Adaptation cost($/ha) $C_{annual}^{crop}$ | Selling price($/tonne) ($P_k$; Eq. 8) |
|---|---|---|---|---|---|---|---|
| **Rice** | 1.50 | 3 | 12 | 6 | 6 | 75 | 729.2 |
| **Maize** | 1.55 | 1.8 | 7.4 | 3.6 | 3.7 | 75 | 299.5 |
| **Sorghum** | 2.13 | 6.8 | 16 | 13.6 | 8 | 75 | 220 |
| **Cassava** | 6.4 | 0.65 | 9.6 | 1.3 | 4.8 | 75 | 207.4 |

*Budget constraints:* In estimating the maximum available budget for adaptation per household, we assumed a household can afford a fixed percentage of disposable income as defined by Kousky and Kunreuther (2014) and further applied in Hudson (2018). However, we assume that farmers can use 6% of their disposable incomes to adapt to damage to houses. When adapting their farms, we assume that farmers can afford up to 50% of their disposable income, as this is an investment in their work. We found these parameters by calibrating the model to surveys conducted in Beira and Nova Sofala that are reported in Duijndam (2024a) and Duijndam et al. (2024b).

### 3.5.2 Migration costs

*Migration decisions:* According to Equation (3), push factors (increasing coastal flood damage and salinity damage, $D_{x,t,i}$) and pull factors (income differentials $Inc_{x,t}$, wealth $W_t$, and amenities $A_x$) interact with mooring factors (fixed migration costs $C_{migration}$) and shape the migration decisions of households in the coastal zone. These factors are calculated as follows: Each node $y$ contains information on income distributions, amenity values, and a distance matrix to all other nodes. The amenity value of node $y$ is a function of the distance to the coast and wealth. We derive the monetary value of these coastal amenities from hedonic pricing studies based on the distance to the coast. We now describe income, migration costs and amenity values in more detail:

*Expected income and migration costs*: For each node $y$ per district, households sample their expected income $Inc_{y,t}$ based on their current position in the log-normal income distribution in the GLOPOP-S database (Ton et al., 2023).

*Migration costs $C_{migration}$* to district $y$ is a function of geographical distance and fixed migration costs (e.g., psychological costs of leaving friends and relatives and moving to an unfamiliar environment). We capture these latter "place attachment costs" with a fixed monetary cost of migration $C_{fixed}$. Ransom (2022) estimates this fixed cost to be between $105,095 and $140,023 for movers in the United States and estimates the total cost of a 500-mile move to be between $394,446 and $459,270. Kennan and Walker (2011) estimate the fixed costs of migration at $312,146 for the average mover in the United States. Based on these figures, we construct a logit function and set the fixed cost of migration (Eq. 10). We assume that the fixed cost of migration is proportional to the cost of housing and thus scale these migration costs to Mozambique price levels using the

differences in housing costs between these countries. Tierolf et al. (2023) use the same logit function for France, with $C_{fixed}$ as EUR 125,000, and Huizinga et al. (2017) provide property costs at the national level. Based on these figures and GDP per capita ratios in 2015 (World Bank, 2015), we scaled the fixed migration cost for Mozambique to $3793, which results in a maximum migration cost of $7586 for very long distances from the coast (Eq. 10).

295
$$C_y^{migration} = \frac{2 * C^{fixed}}{1 + e^{-0.05 * dist_{xy}}}$$
(Eq. 10)

*Amenity value*: We derive the amenity value (scaled to GDP) of living near the coastline based on hedonic pricing studies of coastal property values (e.g. Muriel et al., 2008). However, the coastal amenities for households in Mozambique are based on different values than in similar studies in France and the United States. While in wealthier countries, coastal views increase property values, the data from Mozambique suggest that attractiveness to fisheries is one of the coastal amenities. Therefore,
300   we base our amenity function on Conroy & Milosch (2011) and Muriel et al. (2008) and construct a distance decay function for coastal amenities (Supplementary material S2.2). Households located within 500 m of the coastline experience a coastal amenity premium of 60% of their wealth, which decreases to 3% when located 10 km from the coast. A similar distribution of amenity values as in DYNAMO-M (Tierolf et al., 2023) is applied (Figure S6, Supplementary section) and downscaled based on property values for Mozambique from Huizinga et al. (2017). These estimates perform better than the United States and
305   French estimates, firstly because they account for the dependence of employment on the coast and secondly, because the downscaling with property values captures the income differences between developed and developing countries.

**3.6 Behaviour settings**

The model can also be run for different adaptive behaviour settings. Table 2 shows four settings defined by turning parameter models on and off. First, risk perception $\beta_t$ from Equation 1 can be turned on or off and refers to learning from a flood event
310   (Eq. 5). Higher risk perceptions lead to a higher uptake of adaptation measures. A second parameter is a household's level of awareness of two adaptation measures: (a) the availability of salt-tolerant seeds from seed companies and (b) knowledge about elevating a house to avoid direct flood damage. When these behavioural settings are turned off, as in the 'no adaptation' case (Table 2), agents do not implement adaptation measures and migrate to inland areas (Eqs. 1 and 3). Third, the 'no migration' behaviour setting runs with no resources provided to migrate to inland areas (Eqs. 1 and 2). Finally, the 'full behaviour' setting
315   allows agents to use all options.

Furthermore, these different behavioural settings can be run for different RCP and SSP scenarios (see Supplementary material S1.3). We first simulate the different behaviour settings under a baseline scenario (without future SLR and salt intrusion). Then, the model and behavioural settings are run for two RCP-SSP-coupled scenarios, RCP4.5-SSP2 and RCP8.5-SSP5. Under different climate scenarios, SLR and salt intrusion projections change, while SSP scenarios capture uncertainty in population
320   and income growth (Supplementary S1.2). Income growth changes every year and has a direct influence on input parameters such as property price, adaptation costs, average income, seed costs and producer selling price in the market (Eq. S2, Supplementary S1.1).

**Table 2 Description of different behaviour settings**

| Behaviour setting | Dynamic behaviour to perceive risk | Awareness of adaptation techniques | Migration to inland areas |
|---|---|---|---|
| **Full behaviour** | Yes | Yes | Yes |
| **No adaptation** | Yes | No | Yes |
| **No migration** | Yes | Yes | No |
| **No perception** | No | Yes | Yes |

### 3.7 Calibration and Validation

SLR and associated salt intrusion-induced adaptation and migration are rather new phenomena observed in the last decade, and hence there is not much data on the impact on population dynamics available. Duijndam et al. (2024b) conducted a survey of coastal households (n=828) in the coastal zones of Beira and Nova Sofala in the Sofala province of Mozambique to collect empirical data on the drivers of migration and adaptation under current and future flooding risk (see also Duijndam 2024a). Households were surveyed on their adaptation preferences, willingness to adapt, and existing adaptation. We calibrated our model using empirical data from this survey on current adaptation levels in both households and farms. Table 3 shows the results from the survey relevant to this study. It can be observed that only 38 households out of 413 farming households (9%) have already adapted by using both residential and farm-level adaptation (see "combined").

For validation, we can use the estimate from the survey on the number of households that plan to adapt in the next 5 years, which amounts to an additional 20%. However, self-reported future intentions often fail to translate into real behaviour, described in the literature as the intention- behaviour gap (Kesternich, et al., 2022; Bubeck et al., 2020; Grimmer and Miles, 2017). As a result, we cannot use this number directly as a parameter to validate the model. Considering an intention- behaviour gap of 5-27% (Kesternich, et al., 2022), we estimate that the number of households with both adaptations will rise by about 1-5% in the first 5 years of the model, making a total of 10-14% households in 5 years. As sea-level rise-induced migration increases over the next years and decades, we believe additional longitudinal surveys should be carried out to further improve calibration and validation.

**Table 3 Survey results used for calibrating and validating the ABM. Survey based on interviewing households in Sofala**

| Asset type | Parameter | % of respondents (n=828) |
|---|---|---|
| **Residential** | Adapted with elevating houses | 78.14 |
| | Will adapt in next 5 years | 9.29 |
| | Will not adapt/cannot afford | 12.44 |
| **Farms** | Farming households | 49.87(n=413) |
| | Adapted with crop switch | 11.14 |
| | Will adapt by crop switch in next 5 years | 21.06 |

| | | |
|---|---|---|
| | Will not adapt/cannot afford | 67.31 |
| **Combined** | Adapted both house and farm | 9.20 |
| | Will adapt to both in next 5 years | 19.61 |

## 4 Results

In this section, we present the main results of the model runs for farming households in the coastal flood zone of Mozambique. We first present the results of salt intrusion and asset losses under a full behavioural setting in section 4.1. Section 4.2 shows the results of a single model run, particularly focusing on the exposed population and household adaptations over the model run. Later, in sections 4.3 and 4.4, we present model results under different model settings and adaptation costs, respectively.

**4.1 Salt intrusion and asset losses under full behaviour**

Figure 5 shows the projections of salt intrusion and building losses due to flooding and SLR for farming households in the coastal flood zone (i.e., households with both a house and a farm). The projections are based on 50 Monte Carlo (MC) model runs, with a mean represented by a dark green line and an uncertainty band in light green. We need a sufficiently high number of repetitive runs to capture low frequency flood events in the simulations and to capture their impact. Fifty is a sufficient

number as it can be seen in the convergence test that after 30 runs the standard deviation doesn't change with an additional run (Supplementary 2.3, Figure S6, S7). Panels a, b, and c display the risk of saline intrusion (USD million/year) for different crops under the current climate (Panel a), RCP4.5-SSP2 (Panel b), and RCP8.5-SSP5 (Panel c). The results indicate that coastal farmers in Mozambique face annual damages of US $5 million due to salt intrusion under baseline conditions, increasing to up to US $12.5 million under the extreme climate scenario of RCP8.5. This increase is exponential and is primarily driven by

SLR and the increased frequency of flood events, which deposit large amounts of salt into the soil. Examining Figure 5b (salt intrusion risk projection under RCP4.5), the risk is slightly lower than in the baseline scenario (Figure 5a). This lower risk can be attributed to the growing trend of migration among coastal farmers (Supplementary S1.4, Figure S3d) towards inland locations, which is less observed in the baseline scenario (S1.5, Figure S3e). As farmers migrate, the exposure of farms and their crops decreases, thereby reducing the risk of salt intrusion, despite the natural population growth driven by SSP2 and an

increase in soil salinity. Under the RCP8.5 scenario, however, the increase in risk outweighs the migration flow and the decrease in exposed farms, resulting in a net increase in salt intrusion risk compared to the current climate.

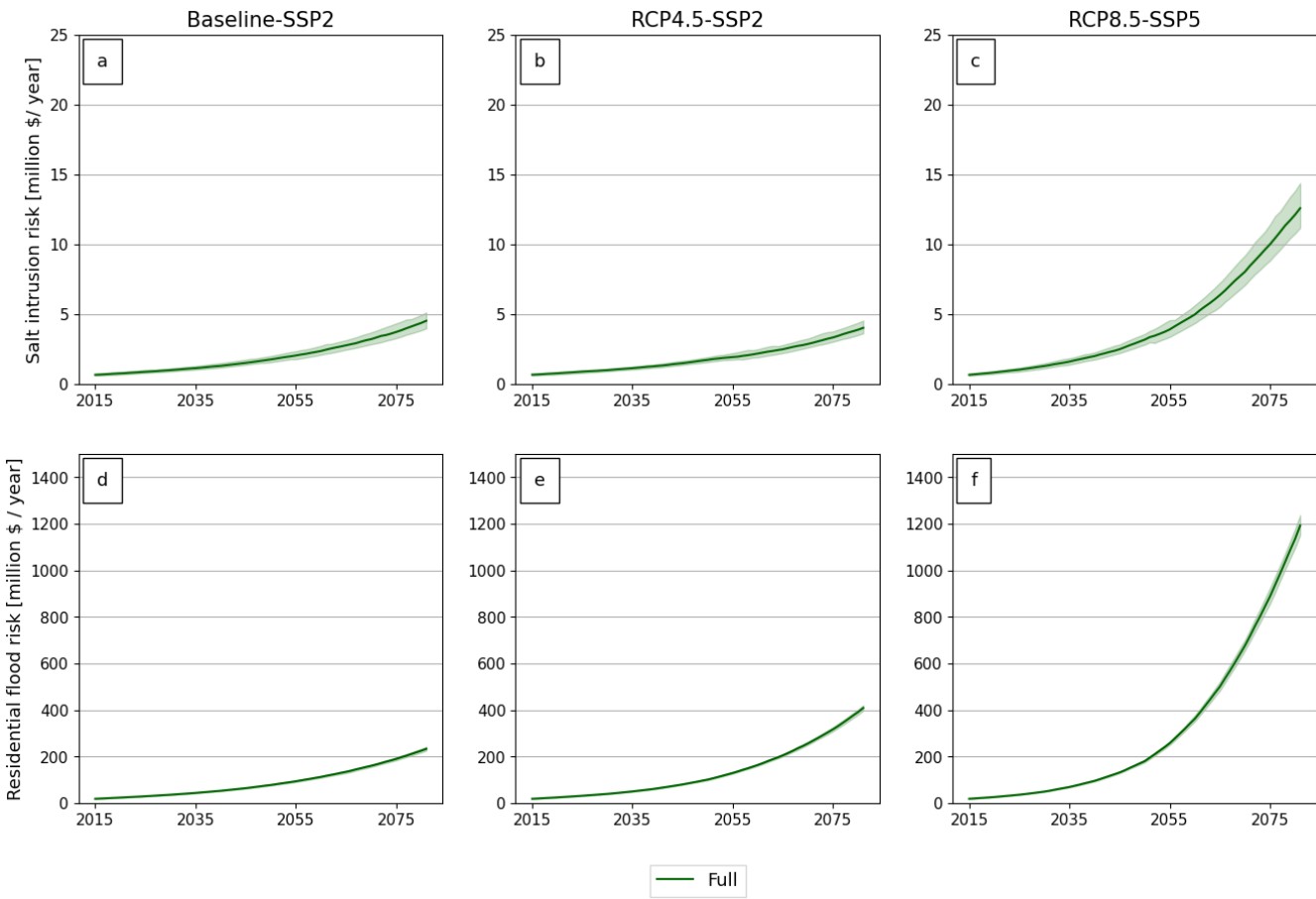

**Figure 5.** Panels a, b and c show the salt intrusion risk (USD million/year) under the current climate, RCP4.5-SSP2 and RCP8.5-SSP5, respectively, under the full behaviour setting (Table 2). Flood risk projections (USD million/year) are shown in panels d, e and f for the current climate, RCP 4.5-SSP2 and RCP 8.5-SSP5, respectively. The green band around the mean line shows the uncertainty in the model due to randomness. Note the shifting *y*-axis.

Flood risks to buildings (USD$ million/yr) with the full behaviour settings (Table 2) are shown in panels d, e, and f in Figure 5 for the current climate, RCP 4.5, and RCP 8.5, respectively. The risk numbers are significantly higher than for the salt intrusion risk, ranging from USD $232 million per year to USD $1200 million per year in 2080 under the current climate and RCP 8.5, respectively. Similar to the salt intrusion risk, flood damage to buildings does not increase as much when comparing the current climate with the RCP 4.5-SSP2 scenario. When a farmer faces salt intrusion damage, 50% of the annual income can be spent on adaptation, which most farmers do and continue to do. However, only 6% of the income can be spent on reducing housing damage, which means under RCP 4.5-SSP2, farmers do not adapt as much and migrate. As migration reduces exposure, the net result is that building damage only increases slightly compared to the current climate. With a GDP of USD $17.8 billion (World Bank 2022), an investment of USD $1212.5 million to cover the loss experienced under RCP 8.5 scenario in year 2080 would be about eight times the current climate funds allocated to Mozambique (~USD $147.3 million; HBS, 2016).

Figure 6.1 (panels a, b, and c) provides more detail on how farmers in the coastal zone experience losses described by expected annual damage (EAD), EAD flood to buildings (x-axis) and EAD crop from salt intrusion (y-axis) for the years 2015, 2050, and 2080, respectively. In addition, the graph also includes a 45-degree line showing farmers who are equally exposed to flood damage to buildings and salt intrusion. Thus, any farmer above the line has a higher risk of salt intrusion, and vice versa. Furthermore, to assess how households with different farm sizes are distributed across these two risk axes, we represent each individual farmer household with a coloured dot depicting farm size (e.g., Esquivel et al., 2021). For the current climate, the building losses per farmer vary from US$0 to $5,500 per year, while salt intrusion losses vary from USD$0 to $1,100 per year. For the year 2080 (RCP4.5-SSP2), these figures range up to $45,000 per year. The values of these damages are quite high compared to the average annual income of farmers in the floodplain ($3,800/yr; Duijndam , 2024). It can be observed that Mozambique does not have many large-scale coastal farmers (large blue dots) and that most large-scale farmers have already adapted by 2015 because, even though they have high losses due to salt intrusion (an annual loss of $750), they also have a high capacity to reduce losses. Thus, the net risk is relatively low, except for some outliers who are closer to the middle-scale farmers in terms of size and wealth. For example, a farmer with a farm area of 5.1 ha would fall into the large (>5 ha) category. However, both the spending capacity (as a function of annual income) and the risk of salt intrusion (as a function of farm area) are similar to those of medium-scale farmers (1–5 ha). In addition, small-scale farmers suffer more damage to buildings from flooding than from salt intrusion. This can be explained by the low adaptation cost to reduce salt risk: cost is a function of farm size, and hence, the cost is relatively low for a small-scale farmer ($150–$1,500). Figure 6.2 shows the same graphs as in Figure 6.1, but farmers are now classified based on their crop type. Sorghum farmers experience little to no salt intrusion damage since sorghum is more salt-tolerant than other crops. This can be derived from the threshold "a" (Equation 7) and the values in Table 1. It can also be observed that cassava farmers experience the most risk (in US$) as compared to any other crops (up to $9,000/yr), which is due to the high yield per hectare of 6.4 t/ha (Table 1). This means the loss per hectare is much larger than for other crops, and the larger income that cassava farmers derive (and thus the higher adaptive capacity) does not offset the losses due to salt intrusion.

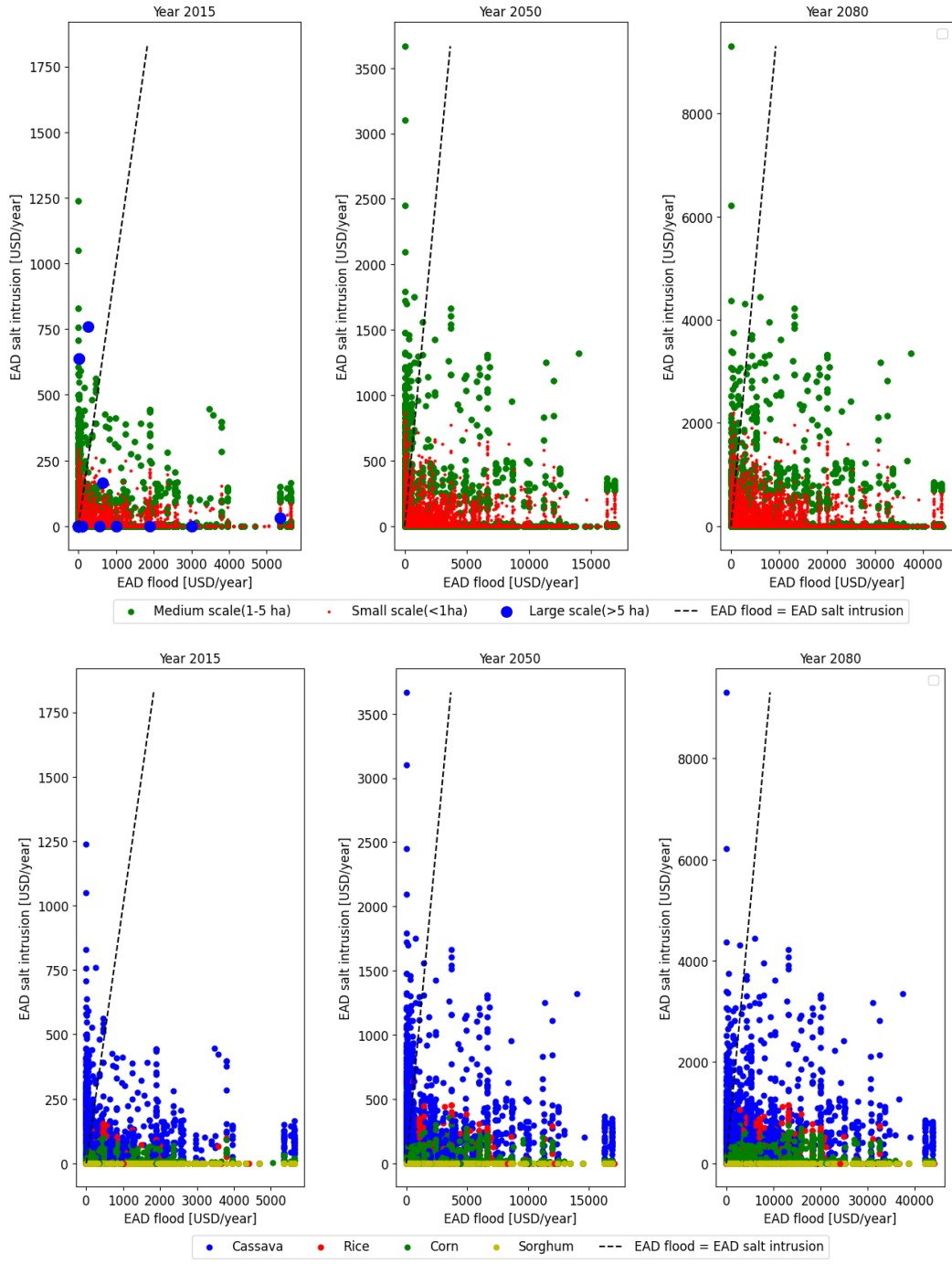

**Figure 6 Salt intrusion risk (*y*-axis) vs. flood risk (*x*-axis) for individual farming households in the Mozambique floodplain under the RCP4.5-SSP2 scenarios (panels a, b and c); panels d, e and f show the same, but now broken down by crop type.**

410 **4.2 Dynamic exposure and adaptation**

To illustrate model behaviour, Figure 7 shows the evolution of the exposed, adapted, and migrated population for a single model run in the province of Sofala (one of the survey locations in Duijndam, 2024). Flood events are represented by vertical dashed lines. Since we assume a flood protection standard with a return period of 10 years for all floodplains, including Sofala, flood events with shorter return periods will not cause impacts. The random simulation of stochastic flood events generates

415 eleven flood events in the province of Sofala by 2080 (Fig. 7). Farming households adapt to flooding and salt intrusion as discussed in section 3.1, resulting in Figure 7b. Another form of adaptation is migration, which can be observed in Figure 7c, where a significant number of households (2032) migrate away from the floodplain, reducing the exposed population.

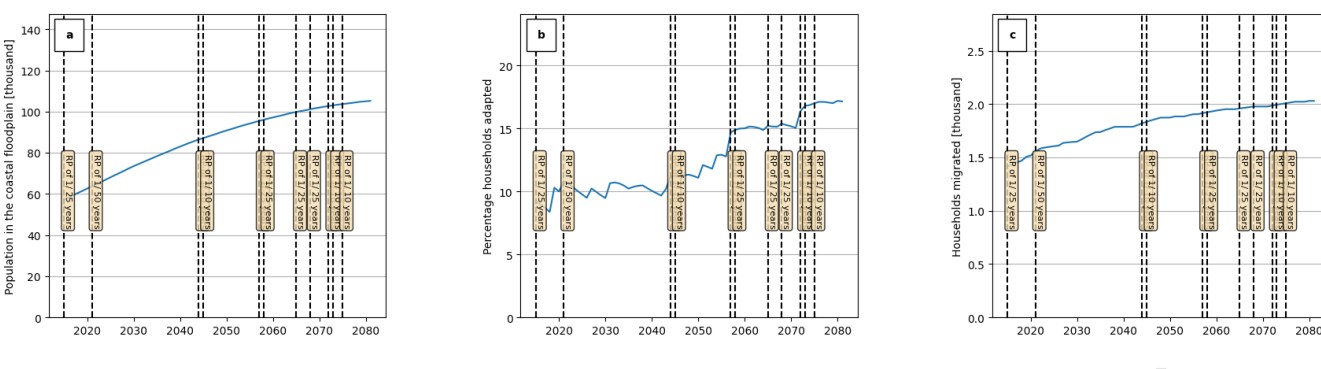

**Figure 7** *a:* Population exposed to the risk of flooding and salt intrusion in the floodplain, *b:* percentage of households adapted, *c:* households
420 migrated. This simulation is made for the coastal province of Sofala under RCP4.5-SSP2 and full behaviour setting.

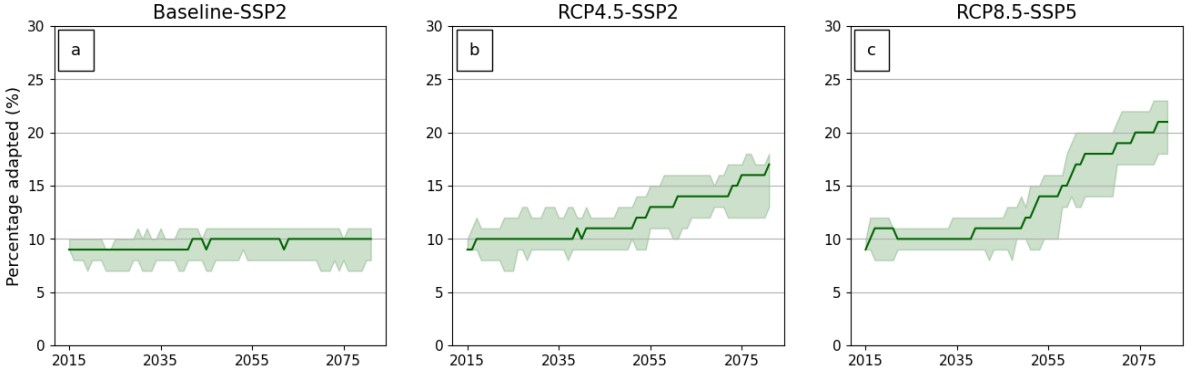

**Figure 8** Percentage of coastal population adapted in Sofala floodplain for 50 MC runs. a) Baseline scenario of no SLR, b) RCP4.5-SSP2, c) RCP8.5-SSP5

Migration increases, even though the percentage of households with adaptation measures increases from 9% at the beginning

425 of the model run to 17% in year 2080 under RCP4.5 and 22% under RCP8.5 (Figure 8b, 8c). The figure shows that at the beginning of the model run, about 9% of the population in the flood zone in the Sofala Province had adapted to flood risk, which is confirmed by empirical data from surveys in the Sofala and Beira areas (Table 3, section 3.7; Duijndam, 2024). By 2020, in 5 years from the model initialization, we observe around 10.1% farming households have adapted under RCP-4.5 and

different MC runs and going up to 11% for some simulations. This simulated behaviour is in line with what would be expected from the (intention-behaviour gap adjusted) survey results in five years' time (10.1-14%, section 3.7). We observe that around two thousand households adapt by migrating to inland areas to reduce SLR and salt intrusion risk (Figure 7c). It can also be observed that people prefer migration over adaptation under a low income situation until 2050, however with GDP growth, households prefer adaptation as place attachment costs (captured by fixed migration cost) offsets the adaptation cost (Figure 7b, 8b, 8c).

## 4.3 Model results under different behaviour settings

To account for stochasticity, Figure 9 shows the average of 50 model runs for coastal Mozambique under the four different behaviour settings (Table 2). Overall, the risk to buildings and crops increases over time, with the number of people in the flood zone gradually increasing. The figure shows that under the 'no perception setting' (red line), households experience the highest salt intrusion and building damage. This is because in the 'no perception setting', households do not change behaviour when at risk, remain in the floodplain, and experience the increasing damage under SLR. The lowest salt intrusion damage is experienced in the 'full behaviour setting', where farmers either fully adapt or migrate and are driven by increased risk perceptions immediately after a flood event. Figure 8 shows that the coastal population in the floodplain is highest under the 'no migration' scenario (~547,000 people under SSP2 and 413,000 people under SSP5) and lowest under the full behaviour scenario (~456,000 people for the RCP4.5-SSP2 scenario and 340,000 people for RCP8.5-SSP5). By 2080, there will be an out-migration of 16.6% for RCP4.5-SSP2 and 17.6% for RCP8.5-SSP5.

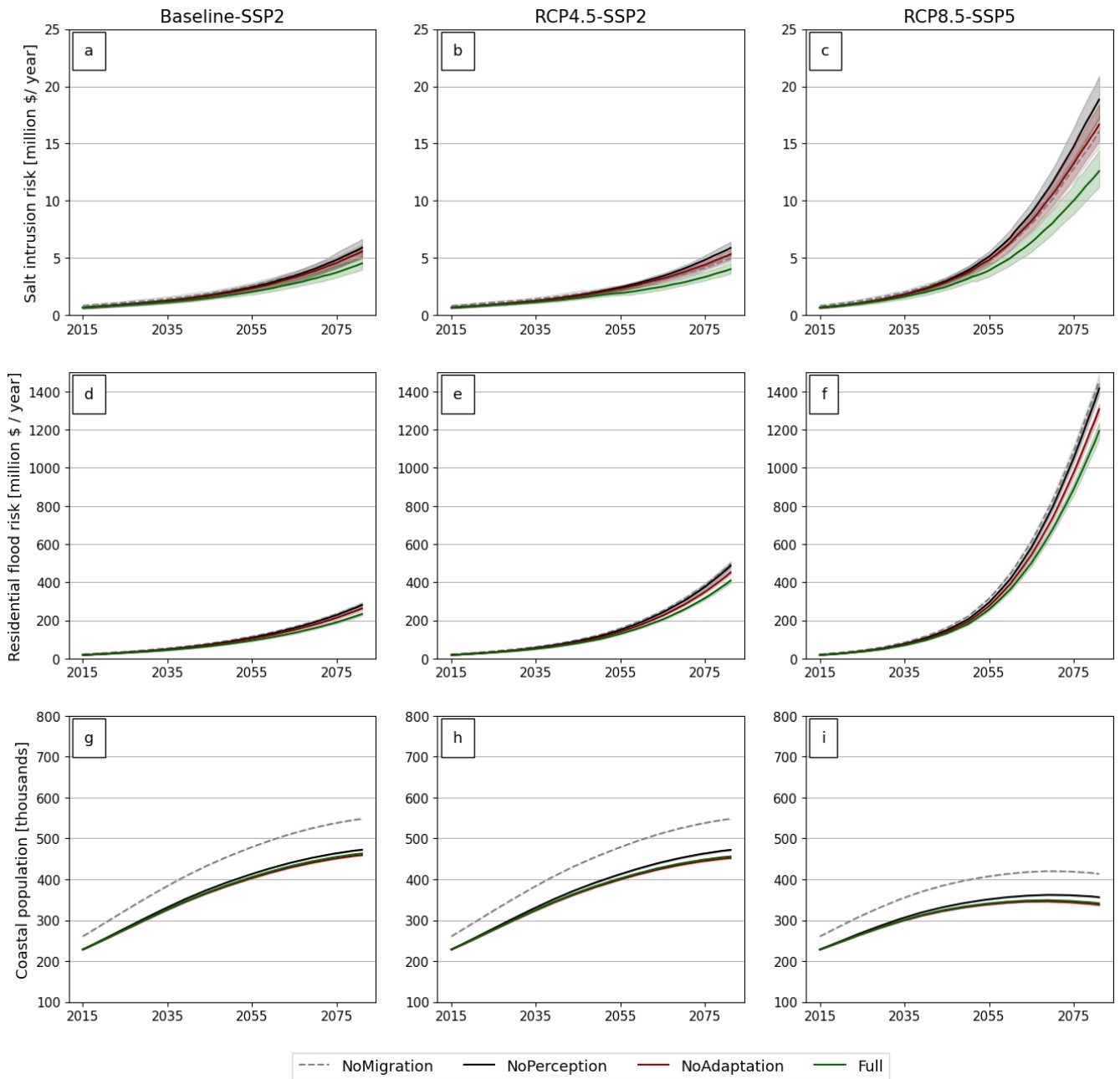

**Figure 9 Losses to farmers from salinization (panels a, b and c), damage to buildings (panels d, e and f), and coastal population (panels g, h, i) under the four different behaviour settings (Table 2) and RCP-SSP scenario combinations.**

Figure 10 shows the percentages of adapted farmers (combined salt and building adaptation) per coastal province under all

450  behaviour settings. Each of these maps shows results assuming RCP4.5 and the 'No migration' and 'Full behaviour' settings. Under the full behaviour settings, the highest percentage of farmers adapt. However, in some provinces, over 65% of the

population does not have the means to adapt because the adaptation costs are too high. The province of Cabo Delgado shows the largest percentage of adapted households (21%), with Maputo city being an outlier at 66% as only 3 farming households were found in 1/100 year flood zone in the year 2080. The percentages of adapted households are higher under the other behavioural settings. For example, under a 'no migration setting', people cannot move away and have only two options: adapt or not adapt. However, some households face financial constraints as only 6% of the annual income can be used for building adaptation and 50% for reducing yield loss and hence these households cannot adapt. Whereas, some richer households who showed migration intentions under full behaviour undergo adaptation under no migration setting(figure 10). Moreover, lowest adaption is observed in Maputo and Nampula province with 15% and 15.8% respectively.

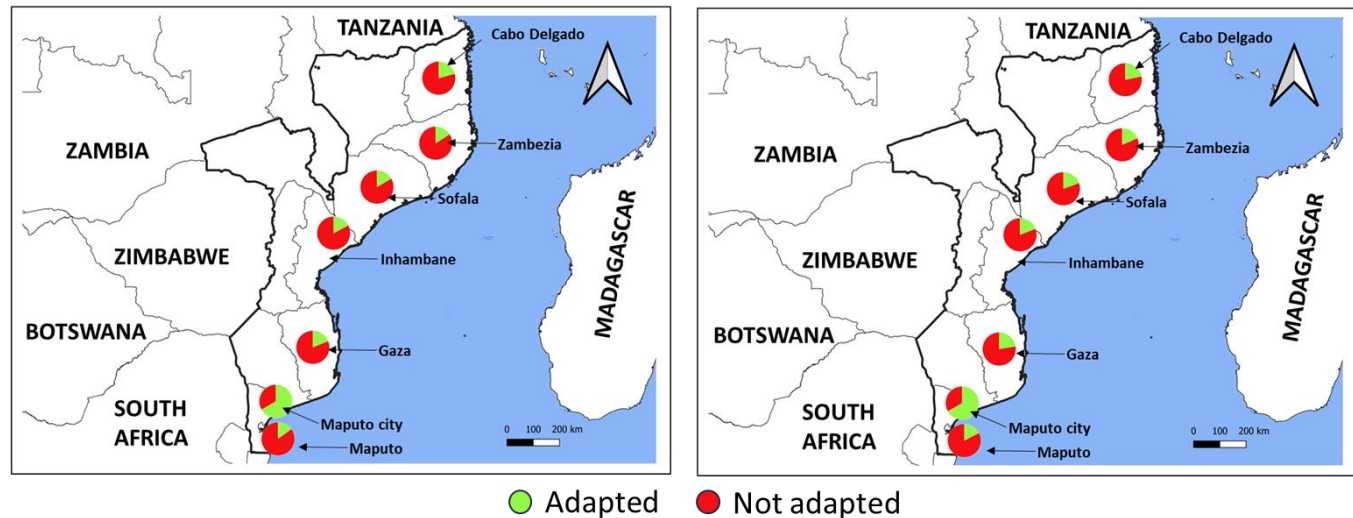

**Figure 10.** The percentage of farmers adapted (to salt intrusion and building damage) in each province under RCP4.5, *left:* full behaviour settings and *right:* no migration

### 4.4 Adaptation cost

Under the influence of increasing building damage and salt intrusion risk, more households adapt and hence annual adaptation costs are projected to increase as well. Figure 11 shows a single model run for the province of Sofala: exposed population (panel a), annual adaptation cost (panel b), and the percentage of households that experience unaffordability (panel c). It can be seen that by 2060, the cumulative adaptation cost in Sofala will rise to 1.1 million USD, further increasing in the next 20 years to 3.1 million dollars in 2080, nearly threefold in two decades (Figure 11b). It can be observed that after every flooding event, the total adaptation cost of the Sofala floodplain (Figure 11c) shows a sudden increase due to more people adapting to SLR and salt intrusion. Moreover, the rate of change of adaptation cost with time also increases, showing exponential growth. It can be observed in figure 10c that 65 percent household in Sofala floodplain cannot afford adaptation because of budget constraints, this is in line with the survey where 67.31 percent households reported they cannot afford adaptation.

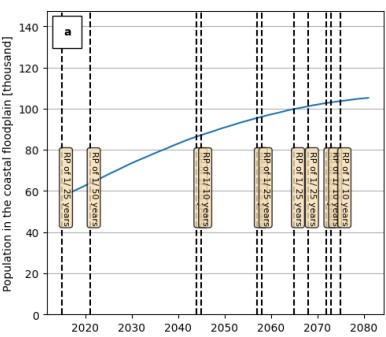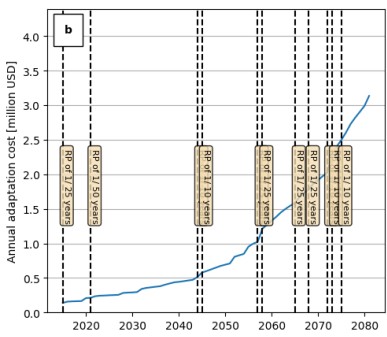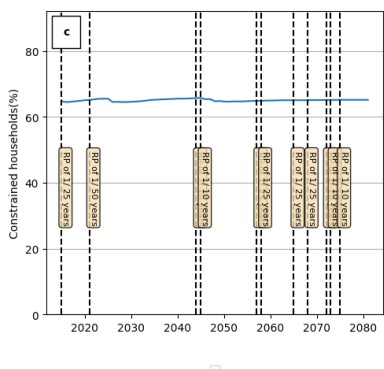

**Figure 11** Left: population in the Sofala floodplain, centre: percentage adapted population, right: adaptation cost. All projections are made for the province of Sofala only.

## 5 Discussion of model sensitivity, limitations and recommendations

### 5.1 Sensitivity analysis

Table 4 summarizes a sensitivity analysis that examines the model's robustness to uncertainties in five model parameters due to their high variability found in the literature. We compare the higher and lower values of these parameters with the standard values of the parameters discussed in the previous sections.

*Expenditure capacity:* In our model, a household can spend up to 6% of their annual income on elevating homes against flooding and up to 50% on switching to a salt-tolerant crop, based on Hudson (2018). Two low and high scenarios of

expenditure capacity (low: 4.8% on homes and 40% on farms; high: 10% on homes and 60% on farms) were studied. Results show little to no sensitivity for controlled baseline scenario of no SLR but sensitivity of the costal population to variations in expenditure capacity under climate scenarios of RCP4.5 and 8.5. For the baseline scenario, when there is no SLR, around 463 thousand households adapt under an expenditure cap of 6% on elevation cost and 50% on crop switch. Under a low scenario of expenditure capacity, the coastal population will decrease 2112 as more farming households prefer migration over

adaptation. On the other hand, under a high expenditure capacity scenario, coastal zone will have 1585 more people as more farming households could afford adaptation.

Under the baseline parameters, most of the households (especially in Inhambane province) are not able to adapt due to unaffordability. The sensitivity to lower expenditure capacity is high but less than comparing differences in people under no SLR and RCP4.5 using standard settings. However, an increase in adaptation capacity (10% on houses and 60% on farms)

enables an increase in households that adapt. This shows that government support, such as adaptation loans or climate funds, could accelerate adaptation and risk reduction.

**Table 4 Sensitivity of the number of coastal population in the year 2080 to five key model parameters under three climate scenarios coupled with socio-economic scenarios. All runs are under full behaviour scenario. Standard parameter values refer to: 6% can be spent on building adaptation, 50% on crop adaptation, the seed cost of 75$/ha, property value of 13370$ in urban, 4457$ in rural, fixed migration cost of 3793 $, salinity projections of ECe2080 are 1.5 times of 2015 for rcp4.5 and 2 times for rcp8.5**

| | | Scenario | | |
|---|---|---|---|---|
| | Parameter setting | Baseline (no SLR) | RCP 4.5- SSP 2 | RCP 8.5 – SSP 5 |
| | | **Population in flood plain in 2080** | | |
| | Standard parameters* | 463017 | 456003 | 341227 |
| **Input parameter** | | **Change from standard population in flood plain** | | |
| **Expenditure capacity (farm %, house %)** | Low (40%, 4%) | +17 | -2112 | -1076 |
| | High (60%, 10%) | +60 | +1585 | +14 |
| **Seed cost** | Low (57.3$/ha) | +2137 | +277 | +50 |
| | High (118$/ha) | -918 | -312 | -53 |
| **Property value** | Low (9626 US$) | -40862 | -41253 | -31378 |
| | High (20456 US$) | +36716 | +38738 | +30125 |
| **Migration cost** | Low (3034 US$) | -2738 | -3089 | -2401 |
| | High (4551 US$) | +3049 | +3136 | +2483 |
| **Salinity projection** | Low (-20%) | -54 | -130 | -89 |
| | High (+20%) | +44 | -277 | +171 |

Seed cost *(crop switch cost):* Adaptation to salt intrusion is simulated, assuming that every farming household buys from the same seed company (Seed Co.) variety SC419 at a homogeneous adaptation cost (75$/ha) (Section 3.5.1). However, in reality, costs vary based on the seed type (SC608: 97$/ha; SC301: 57.3$/ha; SC608: 118$/ha; SSZO, 2019). Based on these alternative values, we tested a low (57.3$/ha) and a high (118$/ha) number. In the baseline scenario, a cost reduction of $57.3/ha results in only 2137 more people compared to a standard crop variety cost of $75/ha. Similar trends are seen in other climate scenarios (RCP 4.5 and RCP 8.5), where a cost decrease led to 277 and 50 more people, respectively. It can be noted that under climate change, seed cost would have less impact on projected coastal population.

*Property value:* The model considers a standard mean property value of $13,370 (Huizinga et al., 2017). However, these numbers are based on real estate data and are highly uncertain with a range given by Huizinga et al. (2017) of -28% up to +53% (-0.28, 0.53). We run two extreme bounds with -28% (9626$) and +53% (20456$). Results show that there can be a difference of more than 40 thousand farmers that can migrate to inland locations. When property values are low, it is easier for households to leave a coastal zone compared to the case of higher property price.

*Migration cost*: We capture migration cost with a "place attachment costs" using a fixed monetary cost of migration $C_{fixed}$ downscaled from Ransom (2022) and applied to net migration cost based on distance (equation 10). However, these values are calculated for migration from New York to Los Angeles, and downscaling based on property value do captures major components, however miss behaviour bias and socio-economic conditions in two countries. Hence, we consider two scenarios of fixed migration cost, a lower cost of $3034 (-20%) and higher cost of $4551 (+20%). We observe that 3089 and 2401 more people would migrate from coastal zone under lower migration costs under RCP4.5-SSP2 and RCP8.5-SSP5 scenario respectively. One the other hand, 3136 less people would migrate under higher migration costs and RCP4.5-SSP2 and 2483 under RCP8.5-SSP5 scenario.

*Soil Salinity projections*: Due to unavailable of soil salinity projections under sea level rise scenarios, we considered them to be similar to projections under climate change (Hassani et al., 2020) and considered that for rcp4.5, an increase of 1.5 times will be observed in the year 2080 compared to 2015. However, to capture the bounded uncertainty we ran two extremes of -20% (1.4 times) and +20% (1.6times) of salinity of 2015. Soil salinity does impact crop damages to the farming households, however in a longer term, by 2080, coastal population shows no to low sensitivity and the behaviour to stay, adapt or migrate is independent.

After running sensitivity analyses for different input variables, it can be seen that the model needs some improvement when upscaled or applied to another coastal plain considering the spatial variability of seed cost, property values and migration cost. For example, the high sensitivity we observed to property values means that regional differences should be accounted for if these are substantial. Moreover, other improvements to the current model could be considering social vulnerability to define adaptation affordability due to its high sensitivity. The heterogeneity of property values needs to be accounted for in a robust analysis, especially in countries with high income inequality. Seed costs vary widely but did not show large differences, which could be due to the fact that house adaptation costs ($1,861) are very high compared to farm adaptation costs ($75/ha), with the average farm size in Mozambique being 1.5 hectares.

## 5.2 Limitations and recommendations

This study was limited by the unavailability of empirical data on salinity levels and projections. We addressed this gap by interpolating the existing global maps from Hassani et al. (2020) into the future. Based on two RCP scenarios, we made two projections of salt intrusion, which can be seen as a sensitivity analysis allowing comparison of future salinity levels with current levels. We project future salt intrusion scenarios assuming a 50% increase in soil salinity for RCP4.5 and 100% for RCP8.5 and neglecting spatial heterogeneity, which is a limitation and source of uncertainty. In addition, the soil salinity simulation does not take into account the spatial heterogeneity of the soil profile for sea salt uptake in coastal soils, which is indeed rather complex. There are two main sources of uncertainty associated with Hassani et al.'s (2020) input map. The first relates to the quantification of error propagation within two processes used by Hassani et al. (2020) to estimate soil salinity: classification and regression. Second, the input soil salinity map is rather coarse (~1 km), which is not directly suitable for farm-level research and is therefore interpolated.

Next to soil salinity, we showed in the sensitivity analysis that the model output (coastal population) is sensitive to the property value and unaffordability. However, there are many other variables that play a role in spending capacity that are not included in our model, such as the psychological cost of investing in adaptation (e.g., Kori, 2023). The effects of income inequality, average household age and gender distribution are not considered, although these factors influence adaptation decisions as they affect social vulnerability. For example, older households face mobility constraints (Cutter et al., 2003), and countries with high income inequality tend to suffer more, as the Gini index is highly correlated with flood fatalities (Lindersson et al., 2023). Meijer et al. (2023) calculated a social vulnerability index to flooding for Madagascar using socio-economic parameters (age, gender, education), and such an approach could be used to improve the realism of the model by defining Gini index to estimate death rate in the model.

Finally, we use household data from GLOPOP-S (Ton et al.,2023), which is aggregated data at the district level. We sampled our agent data from this database. Although in this sampling procedure, we used population density and farm type to place agents on a map, there is considerable uncertainty in assigning agents to a geographic location. Therefore, a more spatially explicit household database could improve the robustness of the model. Moreover, future research should further assess additional behavioural factors following overview studies such as by Noll et al. (2022). For example, such factors can include network effect, worry, climate change beliefs and self-efficacy.

Two main lessons can be drawn from the sensitivity analysis when applying the model to another location or when modelling on a global scale. Firstly, housing prices play a crucial role in estimating damages and modelling adaptation behaviour, with geographical location (rural or urban) and household income serving as essential factors to account for these dynamics. Secondly, adaptation behaviour is strongly influenced by the spending capacity or affordability of the household, with socioeconomic and national poverty line data being used to define affordability (Hudson et al., 2016). Moreover, the current model is limited to constant flood protection standards and a government agent could be modelled which can interact with hazard and coastal households to upgrade flood protection standards.

# 6 Conclusions

SLR will lead to more frequent flooding, and salt intrusion in coastal areas will be a major concern for farming households that are highly dependent on the soil quality for their livelihoods. In this study, we simulated the risk of SLR and flooding to coastal farmers by assessing salt intrusion risk and flood damage to buildings. The results show that the coastal farmers in Mozambique face total losses of $5 million per year under baseline climate scenario (no SLR) and up to $12.5 million per year from salt intrusion and up to $1400 million per year from flooding under RCP8.5 in the year 2080. Sorghum farmers experience little or no damage from salt intrusion, while cassava farmers experience the largest losses, up to $9,000 per year. We show that medium-sized farmers (1–5 ha) face the highest risk because they have large farms but do not have high capacity (i.e. disposable income) to adapt to the increasing risk (Esquivel et al., 2021).

The number of households adapting varies across the province (15%–21%), with salt adaptation being the most adopted because it is the least costly. Despite adaptation measures, of the total of 350,000 farmers in coastal flood zones, about 13%–20% will migrate to safer areas under different settings of adaptive behaviour and different climate scenarios. In some provinces, such as Sofala, the total annual adaptation costs for the farming households will increase to $3.1 million in 2080 with major growth in the last two decades. The paper provides a novel approach to studying the combined effects of SLR and salt intrusion and shows when and where people intend to migration or adapt. It illustrates the importance of considering the heterogeneity in human behaviour in flood impact assessment. Our findings support policymakers in targeting their policies to support individual adaptation, especially in areas of the global south where communities have less coping capacity to deal with SLR. We also show that apart from direct flood risk, salinization in rural areas can have similar impacts on communities. The model could be applied to other countries (and/or the globe) impacted by the combined effect of salt intrusion and SLR by changing the input parameters. Our outcomes open the door for future research and application of the model on a global scale.

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

**Author Contributions**

KP coded and designed the main model, JAdB assisted in coding the main model, JAdB, HdM, WB and JCJHA contributed to the conceptualization and methodology, and KP prepared the manuscript with contributions from all co-authors.

**Funding**

This research has been supported by the H2020 European Research Council (grant no. 884442)

**Acknowledgements**

This research is funded by EU-ERC project COASTMOVE 884442 (www.coastmove.org), and we would like to thank Lars Tierolf, Marijn Ton, Lena Reimann, Toon Haer and others in the COASTMOVE team for their useful discussion.

**Conflict of Interest**

*The authors declare that the research was conducted in the absence of any commercial or financial relationships that could be construed as a potential conflict of interest.*

**Code/Data availability**

The code is freely accessible at https://zenodo.org/records/11194544 and input data can be downloaded from mentioned open sources and pre-processed using scripts provided under directory prepare_input_data.