# Peer review of "Simulating the effects of sea level rise and soil salinization on adaptation and migration decisions in Mozambique"

_EGUsphere, 2024_

## Author Response (AR1)

**RESPONSE to reviewers**

**Simulating the effects of sea level rise and soil salinization on adaptation and migration decisions in Mozambique**

Kushagra Pandey, Jens A. de Bruijn, Hans de Moel, Wouter Botzen, and Jeroen C. J. H. Aerts
* * *
**Reviewer#1**

*I have read the abovementioned paper. The analysis is executed well, and I am generally well satisfied with the paper. However, there are some small issues that should be solved before the paper is accepted. I list them in the order I encountered them in the paper.*

We would like to thank Referee #1 for the kind words and helpful suggestions to improve the paper. We have assessed each comment, and in our reply below, we explain how we would address the comments in our revised manuscript.

*line 66. Gravity model is a statistical, i.e. empirical model. Therefore, writing "statistical and gravity models are less suitable" is confusing for a reader. Authors should explain what other empirical models apart from gravity models are used and reformulate this part.*

The reviewer is correct. We propose changing the text to: "Gravity models are less suitable for individual adaptation and migration decision-making. This is why we selected an agent-based model to simulate these decisions."

*I find motivation for the use of ABM very brief and insufficient. Authors could for example refer to Savin et al. (2023) who argued that ABMs help to combine insights from different disciplines (in this case, behavioural economics/psychology, agriculture and climate adaptation) by seriously addressing the role of bounded rationality, social interaction and agent heterogeneity. You could also refer to a recent literature review on the use of ABM to climate issues like by Castro et al., 2020.*

This is a valuable suggestion. We propose extending the motivation for selecting the agent-based model (ABM) by including the reviewer's suggestion to incorporate Savin et al. (2023) and other relevant literature such as Castro et al. (2020). In the revised paper, we will better highlight the benefits of ABMs for adaptive decision-making by combining insights from different disciplines and addressing aspects like bounded rationality, social interaction, and agent heterogeneity. ABMs are increasingly used for adaptation modeling, as shown by de Ruig et al. (2022) and Haer et al. (2020). Klabunde et al. (2016) have conducted a review study that suggests using different behavior theories within ABMs for migration. This includes the Expected Utility Theory we have included in our model. Thank you for noting this; we will increase the font size accordingly.

Minor comments:

*reference error on p. 6 lie 148*

Correct: we will change this to Fishburn et al. (1981).

*font size in figures like 4 and 5 is too small. Please make sure all figures are readable. Similarly, indices (a,b,c) in figures like 9 and 10 could be improved by putting them outside the plot. Currently they are too small and sometimes overlap with plot content*

Thank you for noting this, we will increase the font size accordingly.

*Finally, the text should be spell checked. e.g. on p. 22 "Third, The input" or "from Ton, Marijn, (2023)"*

Thank you, this should be Ton et al. (2023). We will take extra care to check the references in the final revised version.

**References**

Castro, J., Stefan Drews, Filippos Exadaktylos, Joël Foramitti, Franziska Klein, Théo Konc, Ivan Savin, Jeroen van den Bergh (2020) A review of agent-based modeling of climate-energy policy. WIRES CC, doi.org/10.1002/wcc.647

Haer, T., Husby, T., Botzen, W.J., Aerts, J.C.J.H. (2020). The safe development paradox: an agent-based assessment for flood risk in the European Union. Global Environmental Change, doi.org/10.1016/j.gloenvcha.2019.102009

Klabunde, A., Willekens, F. Decision-Making in Agent-Based Models of Migration: State of the Art and Challenges. Eur J Population 32, 73–97 (2016). https://doi.org/10.1007/s10680-015-9362-0

Ruig, L., Botzen W.J., Haer, T., Brody, S., Czajkowski, de Moel, H., Aerts, J.C.J,.H. (2022) How the U.S. can benefit from risk-based premiums combined with flood protection. Nature Climate Change https://doi.org/10.1038/s41558-022-01501-7 (2022).

Savin, I., Felix Creutzig, Tatiana Filatova, Joël Foramitti, Théo Konc, Leila Niamir, Karolina Safarzynska, Jeroen van den Bergh (2022) Agent-based modeling to integrate elements from different disciplines for ambitious climate policy. WIRES CC, doi.org/10.1002/wcc.811

**Reviewer #2**

*The paper presents a comprehensive and well-constructed model addressing a significant issue, showcasing both the quality of writing and the relevance of the problem tackled. However, to ensure its suitability for publication, it's essential to address several major concerns crucial for enhancing the paper's credibility and impact within the scientific community.*

We would like to thank the Reviewer#2 for the kind words and the helpful suggestions to improve the paper. We agree we can improve the credibility of our paper by improving the validation of our model. We have assessed each comment, and in our reply below, we explain how we would address the comments in our revised manuscript.

Major Comments:

*The paper introduces a model focused on adaptation strategies in rural Mozambique but does not convincingly justify its suitability for this specific environment. The application of expected utility theory seems incongruent with the decision-making realities of rural Mozambican farmers, especially considering insights from urban coastal households (Noll et al., 2022). These insights suggest that adaptation behaviors are heavily influenced by non-economic factors, which are presumably more pronounced in rural settings where traditional knowledge and heuristics play a larger role.*

This is a very relevant comment. The reviewer is right that our model does not capture all behavioral factors that influence adaptation and migration decisions. We have, therefore, extended the EUT model with a risk perception parameter and a risk aversion function (Ruig et al., 2022; Haer et al., 2020; Gandelman, Nestor, 2015). However, other behavioral factors (e.g., worry, fear, social networks etc.) are not included. We propose two ways to better explain the credibility of using the SEUT in our model:

- In the discussion section we will better explain that following the subjective expected utility theory, increase in risk would lead to an increase in adaptation and migration (Reimann et al., 2023; Hauer, M. E. et al., 2020; Chen and Mueller, 2018; Duc Tran et al., 2023). We then show that our model results show a similar increasing trend. However, to validate these results, we will use recent empirical survey results from a survey we conducted in Mozambique with more than 800 respondents. These data show similarly that increase in flood risk will lead to increasing outmigration. We include descriptive statistics from the survey in our revised paper to motivate our behavioural rules and underpin our results.
- Furthermore, we have will make a reference in the Discussion section that future research should further assess additional behavioral factors following overview studies such as by Noll et al. (2022). For example, such factors can include network effect, worry, climate change beliefs and self-efficacy.

*Furthermore, the use of urban housing values to assess expected annual damages further questions the model's applicability, potentially leading to skewed results.*

We agree with the reviewer's comments that urban housing values are generally higher than rural values; hence, we also performed a sensitivity analysis on property values in section 5.1c. A study by the World Bank in 2000 on flooding in Mozambique estimated that construction costs in rural Mozambique are six times lower than those of urban houses, while rural houses are half the size of urban houses. Therefore, for the revised versions, we will reduce the property value and adaptation cost in rural areas by a factor of 3, so that the difference between urban and rural areas is better captured.

*Moreover, the model proposes structural adaptation strategies like elevation that may be impractical and unreflective of the actual measures farmers in rural Mozambique might undertake. This incongruity points to a critical need for the model to integrate a more nuanced understanding of the behavioral patterns and economic realities pertinent to the rural Mozambican context, raising the question of the reliability of the projections presented.*

Before two project colleagues conducted a field survey in Mozambique, we had the same hypotheses as the reviewer. However, our fieldwork in two coastal locations (i.e., Beira and Nova Sofala) in both urban and rural environments revealed that in both areas, elevation is the most selected adaptation option. Hence, the adaptation option 'elevation' is based on the findings in our empirical survey. These data show that elevation does take place (88% of surveyed households have elevation as the first adaptation choice).

To address the reviewers' concern, we will include a new paragraph in the supplementary section of our revised paper on the survey results that substantiate the choices for the adaptation measures included in the model.

*Another significant point that makes me doubt the results presented in this study is the lack of any type of validation. Validation is crucial for establishing the reliability of model outputs, and the absence of any such exercise undermines the strength of the conclusions drawn. Although the challenge of obtaining empirical data for rural Mozambique is acknowledged, the paper would benefit from an attempt to validate the model, perhaps through comparisons with historical trends or alignment with known stylized facts about the region. Without this, the model's predictions remain hypothetical and their relevance to actual scenarios in Mozambique is uncertain.*

The reviewer is correct that validating an ABM in a data poor areas such as Mozambique is quite difficult. This is why most studies use models for exploring options without validation. However, in our study, we have conducted our own survey (*n=828*) to collect empirical data on the drivers of migration and adaptation under current and future flooding risk.

Based on the reviewers commend, we understand we have to more adequately describe our empirical data that underlies our modelling parameters and results. We will, therefore, revise the model using survey data to validate our increase projection of risk and adaptation with survey results. We have calibrated our model using a survey that assesses current adaptation levels in both households and farms. The survey also inquiries about future adaptation plans for the next five years. We will use this information to validate the model's results for the year 2027.

*The sensitivity analysis presented in the study appears to be insufficient. A more robust analysis should consider a wider range of parameter values beyond a few discrete scenarios, which would allow for a better understanding of the model's behavior under different conditions (especially given the points raised before). For instance, the potential impact of soil salinization could be examined over a spectrum of possibilities, such as a 50% increase in salinity levels, to reflect possible variations in climate change outcomes.*

The idea of running 3 values of parameter was to check the upper and lower bound of the possible output to test the uncertainty by running the model at extremes. Good suggestion, we agree with the reviewer to consider the uncertainty in prediction of hazard. We will run additional scenarios, for example, to test the extremes of salinity projections.

*The paper needs to clearly articulate the relationship between income dynamics and environmental factors such as crop salinization. It is important to determine whether farmers' incomes are fixed regardless of the salinization process. The change in crop yield would have a significant impact on farmers' income, which would play a crucial role in their ability to undertake adaptation action or migrate. This is essential for realistically modeling farmers' capacity to adapt or migrate. Maintaining constant income despite climate changes risks underestimating the economic impact of salinization, potentially restricting farmers' options and leading to misleading results.*

Thank you for highlighting this. Upon re-evaluating the paper, we agree indeed that our paper is not clear in describing these relations and will clarify this in a revised version of the manuscript, along the following lines: Our central DEU equations allow for dynamically simulating the relation between income and losses due to salinization and flooded buildings. Thus, our model dynamically simulates the interactions of crop yield loss on income over time and space.

$$DEU_1 = \int_{p_i}^{p_I} \beta_t * pi * U\left(\sum_{t=0}^{T} \frac{W_t + A_{x,t} + Inc_{x,t} - D_{x,t,i}}{(1+r)^t}\right) dp$$

In this equation, $Inc_{x,t}$ is income at a time t in region x which increases over time with the GDP growth rate and $D_{x,t,i}$ is damage experienced by a household every year which has two parts flood damage to buildings + damage to crops due to salt intrusion. The later factor captures the impact of change of crop yield on the ability of the farming household to undertake adaptation decision. In non-flooding years, household income is as expected plus the effect of the GDP growth rate. As an example, we here provide a graph that shows this effect. We propose to include this graph in the revised version of our paper.

[Figure]

*Figure 1 Average household income in Sofala province flood plain under a) GDP growth rate, b) flooding, salt intrusion and no adaptation, c) flooding, salt intrusion and adaptation*

Minor Comments:

*The justification for the choice of 50 repetitions within the model simulations is unclear. I am not saying it is wrong, but such methodological decisions should be supported by a metric that validates the number's adequacy to capture the variability in results.*

Thank you for noting this. We based the 50 repetitions on the work of Tierolf et al. (2023), who also used a maximum of 50 model runs, each having randomly selected variables. Following their work, we will add a new section in the supplementary material with a new figure showing that 50 Monte Carlo runs are sufficient to capture the randomness and low-frequency flood events in the simulations, similar to figure 3 in Tierolf et al. (2024).

*The manuscript has omissions in referencing (reference not found), such as missing citations in figure captions and on page 6, line 145.*

Thank you, we will address these errors and perform an additional check for a revised version.

*The consideration of demographic trends, such as the impact of young people's outmigration on fertility rates and population growth, is lacking. I'm not suggesting that the authors should include it explicitly, but they should reflect upon its implications (Hauer et al., 2024).*

Good suggestion. We propose to address this in the discussion section. Outmigration indeed has an impact on population growth. In our model, population growth is determined by the growth rate and net coastal population. We have already incorporated the impact of out-migration on net coastal population and eventually on population growth. As for the fertility rate, it is important to note that our model considers households rather than individuals and currently does not distinguish between younger/older households. We are continuously developing the model, and we have plans to integrate the average age of households into our calculations. This will inevitably affect the fertility rate, mortality rate, and the social vulnerability index.

*A reflective discussion on the broader implications of the study's results for the scientific literature and their practical significance would be beneficial. The paper should emphasize how its findings contribute new understanding in the field of climate adaptation.*

We will include a discussion on the broader implications of the study and its practical significance, which includes the following points:

- Migration and managed retreat are emerging topics for policy and climate adaptation, especially in areas of the global south where communities have less coping capacity to deal with SLR.
- Our study shows the priority areas for flood and salinization risk and show when and where people intend to migration or adapt.
- The model addresses heterogeneity of households based on socio-economic characteristics such as income differentials, farm size, and types..
- These findings support policymakers in targeting their policies to support individual adaptation.
- We also show that apart from direct flood risk, salinization in rural areas can have similar impacts on communities.

**References:**

Chen, J., Mueller, V. Coastal climate change, soil salinity and human migration in Bangladesh. Nature Clim Change 8, 981–985 (2018). https://doi.org/10.1038/s41558-018-0313-8

Duc Tran, D., Nguyen Duc, T., Park, E., Nguyen Dan, T., Pham Thi Anh, N., Vo Tat, T., & Nguyen Hai, A. (2023). Rural out-migration and the livelihood vulnerability under the intensifying drought and salinity intrusion impacts in the Mekong Delta. International Journal of Disaster Risk Reduction.

Gandelman, Nestor and Hernandez-Murillo, Ruben, Risk Aversion at the Country Level (2015). Available at SSRN: https://ssrn.com/abstract=2646134

Haer, T., Husby, T., Botzen, W.J., Aerts, J.C.J.H. (2020). The safe development paradox: an agent-based assessment for flood risk in the European Union. Global Environmental Change, https://doi.org/10.1016/j.gloenvcha.2019.102009

Hauer, M. E., Fussell, E., Mueller, V., Burkett, M., Call, M., Abel, K., ... & Wrathall, D. (2020). Sea-level rise and human migration. Nature Reviews Earth & Environment, 1(1), 28-39

Hauer, M. E., Jacobs, S. A., & Kulp, S. A. (2024). Climate migration amplifies demographic change and population aging. Proceedings of the National Academy of Sciences of the United States of America, 121(3), e2206192119. https://doi.org/10.1073/PNAS.2206192119/SUPPL_FILE/PNAS.2206192119.SAPP.PDF

Noll, B., Filatova, T., Need, A., & Taberna, A. (2022). Contextualizing cross-national patterns in household climate change adaptation. Nature Climate Change, 12(1), 30–35. https://doi.org/10.1038/s41558-021-01222-3

Reimann, L., Jones, B., Bieker, N. et al. Exploring spatial feedbacks between adaptation policies and internal migration patterns due to sea-level rise. Nat Commun 14, 2630 (2023). https://doi.org/10.1038/s41467-023-38278-y

Ruig, L., Botzen W.J., Haer, T., Brody, S., Czajkowski, de Moel, H., Aerts, J.C.J,.H. (2022) How the U.S. can benefit from risk-based premiums combined with flood protection. Nature Climate Change https://doi.org/10.1038/s41558-022-01501-7 (2022).

Tierolf, L., Haer, T., Athanasiou, P., Luijendijk, A. P., Botzen, W. W., & Aerts, J. C. (2024). Coastal adaptation and migration dynamics under future shoreline changes. Science of the Total Environment, 917, 170239.

Worldbank, 2000. Republic of Mozambique: A Preliminary Assessment of Damage from the Flood and Cyclone Emergency of February-March 2000.

---

## Editor Decision (ED1)

**Supplement to the review of "Simulating the effects of sea level rise and soil salinization on adaptation and migration decisions in Mozambique"**

**Detailed comments**

**Section 3.1 'Adaptation and migration decisions in the 1/100 flood zone':**
Please clarify the presentation of the application of the DEU theory by providing more details on the choice and definition of certain functions and parameters, along with a more critical discussion of the methodology chosen for modeling adaptation and migration decisions, following the suggestions below.

- **Redistribution of information presented in the supplementary Section S1.1**: the supplementary information is important to understand the model and it should be moved to the main paper (Section 3.1), as it would help clarify how the utility function U and the risk perception parameter $\beta$ are defined; these are critical elements that should be included in Section 3.1, to ensure that the characterization of the risk aversion and perception is more clear.
- **Formulation of the Discounted Expected Utility (DEU) equations**: the chosen formulation of DEU (Eq. 1-3) raises some questions regarding the use of the sum of all discounted economic terms over time as argument of the utility function, rather than summing the discounted utilities (U values) themselves over time, as I think it is often done in the literature (e.g., Coble and Lusk, 2010); I believe that some compensations of economic terms (income and costs) occurring at different times are possible in the current formulation. While the chosen formulation may align with the DEU theory, it is worth considering and discussing whether it properly captures the agents' preferences related to the temporal distribution of wealth, income, damages, and costs over time (within T). It would be beneficial to check and discuss this choice (and its possible advantages or limitations) in greater detail in the text, to enhance the clarity of the DEU formulation and assumptions (regarding the agents' time and risk preferences) and to strengthen the rationale of the approach regarding the disentanglement of time preferences.
- **Discount factor choice and suitability for rural Mozambique**: it would be good to clarify how the value (r=3.2%) has been chosen and discuss how it can reflect the time preferences of households in the case study of Mozambique, as the citation reported (Evans and Sezer, 2005) should refer to the European Union context.
- **Definition of all parameters and constants used in all equations**: the definition of a few parameters is missing, and should be included more clearly and explicitly within the paper for the sake of clarity (not referring to possible references only); in the current version, the values and meaning of a few parameters are not reported, i.e., pi, p1 and T in Eq. 1-3 (at the moment, one can understand or guess their meaning or value, e.g. T should be 15 years, see L. 167); the terms c, d and sigma ($\sigma$) in Eq. S1-S2 are not defined.

**References**:

- Coble, K.H.; Lusk, J.L (2010). At the nexus of risk and time preferences: An experimental investigation. J. Risk Uncertain, 41, 67–79. https://doi.org/10.1007/s11166-010-9096-7
- Evans, D. J., & Sezer, H. (2005). Social discount rates for member countries of the European Union. Journal of Economic Studies, 32(1), 47–59. https://doi.org/10.1108/01443580510574832
* * *
**Technical corrections**

- L. 123-126 (Section 2, Case study): in addition to internal migration within Mozambique, the issue of out-migration flows to other countries, mainly to other southern African countries, should be mentioned here to provide a more complete context (maybe this could be incorporated after the sentence: *"Internal socioeconomic-driven migration has already been an issue in Mozambique since the 1980s (First, 1983) … .")*
- L. 135: the word 'risk' may be missing in the sentence 'reduce soil salinity (risk) on their farmland by switching to a more salt-tolerant variety
- L. 172.173 (Section 3.2): clarify what is meant by 'exclude higher return periods from our analysis'
- L. 179: the wording with 'selecting' does not seem to be the most appropriate in the sentence 'Synthetic future flood events are simulated by randomly selecting for each administrative unit and the exceedance probability of each flood event' (maybe the object complement after selecting should be explicited, e.g. 'by randomly selecting events')
- Missing definitions of acronyms, check and clarify all abbreviations and special notations (e.g., ECe, line 37 of introduction and Figure 2; EAD in Figure 6 labels; t+=1 should be t=t+1, in Figure 2, avoiding special informatic notation)
- L. 151: I think that 'per county' should be 'per district' in the Mozambique context
- L. 348: In the sentence 'We first present the results of salt intrusion and asset losses under a full behavioural setting.', it would be good to specify in Section 4.1 for consistency with the rest.
- L. 391: The sentence "With a GDP of USD $17.8 billion (World Bank 2022), an investment of USD $1212.5 million to cover the loss would be …" can be better linked and clarified in the context of the results presented (e.g., specifying that $1212.5 is the annual loss expected by 2080 in the RCP8.5 scenario; moreover, for consistency, I would suggest reporting the same number ($1212.5) at L. 385.
- L. 478-480: these sentences can be improved and clarified ("However, some households face financial constraints as only 6% of the annual income can be used for building adaptation and 50% for reducing yield loss and cannot adapt. Whereas, some richer households who showed migration intentions under full behaviour shows adaptation.")
- L. 480: typo in 'Thew lowest'
- L. 500-501: this sentence can be improved ("It can be observed in figure 10c that 65 percent household in Sofala floodplain cannot afford adaptation because of

budget constraints, this was also observed in the survey where 67.31 percent households reported they cannot afford adaptation."), e.g., maybe '... which is in line with the survey...'

- L. 600: I would suggest that 'unaffordability of adaptation' would read better
- L.625-626: "The results show that the coastal farmers in Mozambique face total losses of $5 million per year under baseline climate scenario ..." – it would be good to remind here (in a parenthesis maybe) that this scenario refers to no SLR
- Section 5.2: a sentence could be added here to remind the limitation of considering a single fixed flood protection standard (with a return period of 10 years), as this seems an important simplification of possible spatial and temporal changes in the coastal protection standards that may influence the uncertainty of the results.
- Table 3: I would suggest reporting the farming households percentage in the Table caption or in a separate line (for consistency with the other table entries reporting percentages of households that have adapted or indent to); also, in the first line, consider adding the word 'houses' in 'Adapted with elevating houses'
- Table 4: the caption can be improved, e.g. "Sensitivity of the number of coastal population in the year 2080 to five key model parameters ... "
- Reference lists in both the main paper and Supplementary material, check and proof-read, e.g. remove the double entry for Schiavina et al. (2019) from the SI list; complete the information missing for Duijndam, S. J. (2024) (Floods of movement: Drivers of human migration under sea-level rise and flood risk. [PhD-Thesis - Research and graduation internal, Vrije Universiteit Amsterdam]. https://doi.org/10.5463/thesis.705)
- Check that the references to figures in the Supplementary material are clear with continuous consistent numbering and format within the Supplementary material (e.g., Figures S1 to S8), i.e., probably Figures 12 and 13 should appear as Figures S7 and S8

---

## Author Response (AR2)

1. ***Redistribution of information presented in the supplementary Section S1.1:*** *the supplementary information is important to understand the model and it should be moved to the main paper (Section 3.1), as it would help clarify how the utility function U and the risk perception parameter β are defined; these are critical elements that should be included in Section 3.1, to ensure that the characterization of the risk aversion and perception is more clear.*

We moved supplementary 1.1 to main text and adjusted the indexing of sections and equations accordingly.

2. ***"Formulation of the Discounted Expected Utility (DEU) equations:*** *the chosen formulation of DEU (Eq. 1-3) raises some questions regarding the use of the sum of all discounted economic terms over time as argument of the utility function, rather than summing the discounted utilities (U values) themselves over time, as I think it is often done in the literature (e.g., Coble and Lusk, 2010); I believe that some compensations of economic terms (income and costs) occurring at different times are possible in the current formulation. While the chosen formulation may align with the DEU theory, it is worth considering and discussing whether it properly captures the agents' preferences related to the temporal distribution of wealth, income, damages, and costs over time (within T). It would be beneficial to check and discuss this choice (and its possible advantages or limitations) in greater detail in the text, to enhance the clarity of the DEU formulation and assumptions (regarding the agents' time and risk preferences) and to strengthen the rationale of the approach regarding the disentanglement of time preferences."*

We added the following Footnote 1 in Section 3.1 to discuss this point:

"Our formulation of the discounted expected utility functions includes a summation of monetary terms that occur over time as is line with related ABM applications (e.g. Haer et al., 2019; de Ruig et al., 2022; Tierolf et al., 2023), instead of a summation of discounted utility values themselves over time (Coble and Lusk, 2010). Our approach is consistent with the use of a time discount rate estimated for monetary values instead of a utility discount rate, but may be a simplification for capturing agents' preferences related to the temporal distribution of the included monetary amounts over time. Although we do not have data on such preferences for Mozambique to directly tests for this, the model calibration and validation exercises show that our behavioural rules adequately predict observed adaptation decisions in Mozambique (see sections 3.7 and 4.2). This gives confidence in our approach."

3. ***"Discount factor choice and suitability for rural Mozambique***: *it would be good to clarify how the value (r=3.2%) has been chosen and discuss how it can reflect the time preferences of households in the case study of Mozambique, as the citation reported (Evans and Sezer, 2005) should refer to the European Union context."*

We added the following Footnote 2 in Section 3.1 to clarify this choice:

"This value of the time discount rate is based on estimates derived from the European context, since a Mozambique estimate is lacking. One could expect that the actual discount rate in Mozambique is higher than this value, resulting in a too high weight given to monetary values in the far future. However, such an effect is counteracted by our choice for a relatively short time horizon of 15 years over which future values are included in the utility calculation. Our model calibration and validated analyses demonstrate that our combined choice of behavioural parameters performs well, in a sense that modelled adaptation outcomes match those observed in Mozambique with survey data (see sections 3.7 and 4.2)."

4. ***Definition of all parameters and constants used in all equations***: *the definition of a few parameters is missing, and should be included more clearly and explicitly within the paper for the sake of clarity (not referring to possible references only); in the current version, the values*

*and meaning of a few parameters are not reported, i.e., pi, p1 and T in Eq. 1-3 (at the moment, one can understand or guess their meaning or value, e.g. T should be 15 years, see L. 167); the terms c, d and sigma ($\sigma$) in Eq. S1-S2 are not defined.*

We edited the document by adding definitions of parameters and terms.

**Technical corrections**

1. *L. 123-126 (Section 2, Case study): in addition to internal migration within Mozambique, the issue of out-migration flows to other countries, mainly to other southern African countries, should be mentioned here to provide a more complete context (maybe this could be incorporated after the sentence: "Internal socioeconomic-driven migration has already been an issue in Mozambique since the 1980s (First, 1983) … .")*

Added apart from out-migration to other south African countries (Facchini et al., 2013)

2. *L. 135: the word 'risk' may be missing in the sentence 'reduce soil salinity (risk) on their farmland by switching to a more salt-tolerant variety*

Corrected

3. *L. 172.173 (Section 3.2): clarify what is meant by 'exclude higher return periods from our analysis'*

Mentioned the return periords (1/2 years and 1/5 years)

4. *L. 179: the wording with 'selecting' does not seem to be the most appropriate in the sentence 'Synthetic future flood events are simulated by randomly selecting for each administrative unit and the exceedance probability of each flood event' (maybe the object complement after selecting should be explicited, e.g. 'by randomly selecting events')*

Added 'an event type (by return-period)' to clarify

5. *Missing definitions of acronyms, check and clarify all abbreviations and special notations (e.g., ECe, line 37 of introduction and Figure 2; EAD in Figure 6 labels; t+=1 should be t=t+1, in Figure 2, avoiding special informatic notation)*

Corrected the figure and defined EAD as Expected Annual Damage in the text

6. *L. 151: I think that 'per county' should be 'per district' in the Mozambique context*

Right, modified the word

7. *L. 348: In the sentence 'We first present the results of salt intrusion and asset losses under a full behavioural setting.', it would be good to specify in Section 4.1 for consistency with the rest.*

Mentioned the section index

8. *L. 391: The sentence "With a GDP of USD $17.8 billion (World Bank 2022), an investment of USD $1212.5 million to cover the loss would be …" can be better linked and clarified in the context of the results presented (e.g., specifying that $1212.5 is the annual loss expected by 2080 in the RCP8.5 scenario; moreover, for consistency, I would suggest reporting the same number ($1212.5) at L. 385.*

Added a sentence 'experienced under RCP 8.5 scenario in year 2080'

9. *L. 478-480: these sentences can be improved and clarified ("However, some households face financial constraints as only 6% of the annual income can be used for building adaptation and 50% for reducing yield loss and cannot adapt. Whereas, some richer households who showed migration intentions under full behaviour shows adaptation.")*

Modified the sentence for clarification

10. *L. 480: typo in 'Thew lowest'*

Modified to 'Moreover, lowest'

11. *L. 500-501: this sentence can be improved ("It can be observed in figure 10c that 65 percent household in Sofala floodplain cannot afford adaptation because of budget constraints, this was also observed in the survey where 67.31 percent households reported they cannot afford adaptation."), e.g., maybe '… which is in line with the survey…'*

Modified, now it flows better, thanks

12. *L. 600: I would suggest that 'unaffordability of adaptation' would read better*

We previously define 'spending capacity or affordability' as a characteristic of a household and would like to stick with that

13. *L.625-626: "The results show that the coastal farmers in Mozambique face total losses of $5 million per year under baseline climate scenario …" – it would be good to remind here (in a parenthesis maybe) that this scenario refers to no SLR*

Right, added in the text

14. *Section 5.2: a sentence could be added here to remind the limitation of considering a single fixed flood protection standard (with a return period of 10 years), as this seems an important simplification of possible spatial and temporal changes in the coastal protection standards that may influence the uncertainty of the results*

Added a sentence 'Moreover, the current model is limited to constant flood protection standards and a government agent could be modelled which can interact with hazard and coastal households to upgrade flood protection standards.' to define limitation of the model and room for further improvement.

15. *Table 3: I would suggest reporting the farming households percentage in the Table caption or in a separate line (for consistency with the other table entries reporting percentages of households that have adapted or indent to); also, in the first line, consider adding the word 'houses' in 'Adapted with elevating houses'*

Changed to Adapted with elevating houses

16. *Table 4: the caption can be improved, e.g. "Sensitivity of the number of coastal population in the year 2080 to five key model parameters … "*

Modified the caption to 'Sensitivity of the number of coastal population in the year 2080 to five key model parameters under three climate scenarios coupled with socio-economic scenarios.

17. *Reference lists in both the main paper and Supplementary material, check and proof-read, e.g. remove the double entry for Schiavina et al. (2019) from the SI list; complete the information missing for Duijndam, S. J. (2024) (Floods of movement: Drivers of human migration under sea-level rise and flood risk. [PhD-Thesis - Research and graduation internal, Vrije Universiteit Amsterdam]. https://doi.org/10.5463/thesis.705)*

Completed the reference

18. *Check that the references to figures in the Supplementary material are clear with continuous consistent numbering and format within the Supplementary material (e.g., Figures S1 to S8), i.e., probably Figures 12 and 13 should appear as Figures S7 and S8*

Checked and modified

References:

Coble, K. H., & Lusk, J. L. (2010). At the nexus of risk and time preferences: An experimental investigation. *Journal of Risk and Uncertainty*, *41*, 67-79.

De Ruig, L.T., Haer, T., de Moel, H., Brody, S.M., Botzen, W.J.W., Czajkowski, J., Aerts, J.C.J.H. (2022). How the U.S. can benefit from risk-based premiums combined with flood protection. *Nature Climate Change,* 12: 995-998*.*

Facchini G, Mayda AM, Mendola M (2013) South-South migration and the labour market: evidence from South Africa. J Econ Lit:1–24

Haer, T., Botzen, W.J.W., Aerts, J.C.J.H. (2019). Advancing disaster policies by integrating dynamic adaptive behaviour in risk assessments using an agent-based modelling approach. *Environmental Research Letters*, 14:4.